# Improved representation of plant physiology in the JULES-vn5.6 land surface model: Photosynthesis, stomatal conductance and thermal acclimation

Rebecca J. Oliver[1], Lina M. Mercado[1,2], Doug B. Clark[1], Chris Huntingford[1], Christopher M. Taylor[1,5], Pier Luigi Vidale[3], Patrick C. McGuire[3], Markus Todt[3], Sonja Folwell[1], Valiyaveetil Shamsudheen Semeena[1], Belinda E. Medlyn[4]

[1] UK Centre for Ecology and Hydrology, Wallingford, OX10 8BB, UK

[2] College of Life and Environmental Sciences, University of Exeter, Exeter, EX4 4RJ, UK

[3] Department of Meteorology and National Centre for Atmospheric Science, Reading University, Reading RG6 6BB, UK

[4] Hawkesbury Institute for the Environment, Western Sydney University, Australia

[5] National Centre for Earth Observation, Wallingford, OX10 8BB, UK

Journal: GMD – Development and technical paper

*Correspondence to:* R. J. Oliver (rfu@ceh.ac.uk)

**Abstract.**

Carbon and water cycle dynamics of vegetation are controlled primarily by photosynthesis and stomatal conductance ($g_s$). Our goal is to improve the representation of these key physiological processes within the JULES land surface model, with a particular focus on refining the temperature sensitivity of photosynthesis, impacting modelled carbon, energy and water fluxes. We test (1) an implementation of the Farquhar et al. (1980) photosynthesis scheme and associated plant functional type-dependent photosynthetic temperature response functions, (2) the optimality-based $g_s$ scheme from Medlyn et al. (2011), and (3) the Kattge and Knorr (2007) photosynthetic capacity thermal acclimation scheme. New parameters for each model configuration are adopted from recent large observational datasets that synthesise global experimental data. These developments to JULES incorporate current physiological understanding of vegetation behaviour into the model, and enable users to derive direct links between model parameters and on-going measurement campaigns that refine such parameter values. Replacement of the original Collatz *et al.* (1991) C$_3$ photosynthesis model with the Farquhar scheme results in large changes in GPP for current-day, with ~10% reduction in seasonal (June-August; JJA and December-February; DJF) mean GPP in tropical forests, and ~20% increase in the northern high latitude forests in JJA. The optimality-based $g_s$ model decreases the latent heat flux for the present-day (~10%, with an associated increase in sensible heat flux) across regions dominated by needleleaf evergreen forest in the northern hemisphere summer. Thermal acclimation of photosynthesis coupled with the Medlyn $g_s$ scheme reduced tropical forest GPP by up to 5%, and increased GPP in the high northern latitude forests by between 2 to 5%. Evaluation of simulated carbon and water fluxes by each model configuration against global data products show this latter configuration generates improvements in these key areas. Thermal acclimation of photosynthesis coupled with the Medlyn $g_s$ scheme improved modelled carbon fluxes in tropical and high northern latitude forests in JJA, and improved the simulation of evapotranspiration across much of the northern hemisphere in JJA. Having established good model performance for the contemporary period, we force this new version of JULES offline with a future climate scenario corresponding to rising atmospheric greenhouse gases (SSP5 RCP8.5). In particular, these calculations allow understanding of the effects of long-term warming. We find that the impact of thermal acclimation coupled with the optimality-based $g_s$ model on simulated fluxes increases latent heat flux (+50%) by year 2050 compared to the JULES model configuration without acclimation. This new JULES configuration also projects increased GPP across tropical (+10%) and northern latitude regions (+30%) by 2050. We conclude that thermal acclimation of photosynthesis with the Farquhar photosynthesis scheme and the new optimality-based $g_s$ scheme together improve the simulation of carbon and water fluxes for current-day, and has a large impact on modelled future carbon cycle dynamics in a warming world.

## 1. Introduction

Photosynthesis and stomatal conductance ($g_s$) together exert a strong control over the exchange of carbon, water and energy between the land surface and the atmosphere. The behaviour of stomatal pores on the leaf surface link these processes, controlling the amount of carbon dioxide ($CO_2$) entering, and water leaving each leaf. Photosynthesis represents the largest exchange of carbon between the land and atmosphere (Friedlingstein et al., 2020), being more substantial than respiration loss. This imbalance is central to the global carbon cycle because it slows the rate of accumulation of $CO_2$ in the atmosphere caused by fossil fuel burning, and therefore also lowers the rate of atmospheric temperature increase. As stomata open to take up $CO_2$ for photosynthesis, plants also lose water through transpiration, and this flux has been estimated to account for 60–80% of evapotranspiration (ET) across the land surface (Jasechko et al., 2013; Schlesinger and Jasechko, 2014). Hence, for vegetated surfaces, transpiration is the primary driver of the latent heat flux (LE), the latter describing the overall transfer of water vapour to the atmosphere. The partitioning of available net radiation between LE and sensible heat (H) is also a key determinant of land surface temperature, therefore having a feedback on photosynthesis and other key metabolic processes that influence the global carbon cycle such as plant respiration.

Land surface models (LSMs) simulate the exchange of carbon, water and energy between the land surface and the atmosphere, providing the lower boundary conditions for the atmospheric component of Earth System Models (ESMs) when run in a coupled configuration. ESM projections form the main tool to predict future climate change and underpin much of the regular United Nations Intergovernmental Panel on Climate Change (IPCC) reports that inform policymakers. However, ESM predictions of the global carbon sink are fraught with large uncertainties surrounding projections of future carbon uptake (Friedlingstein et al., 2014), causing uncertainty in any translation from $CO_2$ emissions to atmospheric $CO_2$ trajectory. A lack of knowledge in how the global carbon cycle operates creates uncertainties in translating from emissions to global warming, and these uncertainties are a sizeable fraction of those associated with unknowns of physical climate processes (Huntingford et al., 2009). Therefore, given the critical role of both photosynthesis and $g_s$ in determining land-atmosphere exchanges, their accurate representation and parameterisation in LSMs is of paramount importance. Booth et al. (2012) show that a significant uncertainty is the temperature sensitivity of photosynthesis, and suggest that thermal acclimation of photosynthesis – where plants adjust their optimum temperature for photosynthesis to growth conditions experienced over the timescale of days to weeks - might reduce the spread in modelled carbon exchange. Yet despite strong evidence of the thermal acclimation capability of plant photosynthesis (Dusenge et al., 2020; Slot et al., 2021; Way et al., 2017; Way and Yamori, 2014; Yamaguchi et al., 2016), incorporation of this process in large-scale LSMs is limited to only a few e.g. TEM (Chen and Zhuang, 2013), CLM4.5 (Lombardozzi et al., 2015), LM3 (Smith et al., 2016), JULES (Mercado et al., 2018), ORCHIDEE (Krinner et al., 2005) and BETHY (Ziehn et al., 2011), and is not yet commonly represented in ESMs. Currently, the majority of LSMs and ESMs use simple fixed (i.e. non-acclimating) temperature response functions for photosynthetic capacity parameters (Smith and Dukes, 2013), which, in general, cause the rate of leaf photosynthesis to increase with temperature to an optimum and then decrease under higher temperatures. These functional forms are either generic for all $C_3$/$C_4$ species and fixed in time and space, or are dependent on a small number of plant functional types (PFTs) but again fixed in time and space. Consequently, climate-carbon feedbacks in ESMs are sensitive to the assumed value of the fixed optimum temperature for photosynthetic capacity ($T_{opt}$), because the amount of carbon assimilated

depends on whether leaf temperature is dominantly above or below $T_{opt}$. Improved process representation of $g_s$,
photosynthesis, and its temperature sensitivity in LSMs is necessary to support robust predictions of global climate
change via their coupling into ESMs. Modelling studies have shown how photosynthesis and $g_s$ impact climate
feedbacks, play a critical role in how climate will change, and strongly influence climate-induced impacts such as
water resources (Betts et al., 2007; Cruz et al., 2010; De Arellano et al., 2012; Gedney et al., 2006; Kooperman et
al., 2018; Zeng et al., 2017).
This study, therefore, updates the plant physiology routines in the Joint UK Land Surface Environment Simulator
(JULES-vn5.6) LSM, the land-surface component of the UK Hadley Centre ESM (Sellar et al., 2019). To date,
JULES has employed the mechanistic $C_3$ photosynthesis scheme of Collatz et al. (1991) ("Collatz"). However,
the Farquhar et al. (1980) ("Farquhar") scheme is more generally adopted by those modelling photosynthetic
response and by researchers analysing data from empirical studies. The Farquhar scheme has been recently
implemented in JULES by Mercado et al. (2018) for $C_3$ plant types, albeit using a big leaf canopy scaling approach
and was not parameterised and evaluated for global applications. Here we build on that previous study by using a
data-driven approach incorporating data from multiple biomes to parameterise the Farquhar model photosynthetic
capacity parameters and their temperature sensitivity so it is amenable for use in global studies. Our specific
rationale for including the Farquhar photosynthesis scheme is twofold. Firstly, studies by Rogers et al. (2017) and
Walker et al. (2021) demonstrate that despite only the Collatz or Farquhar descriptions of leaf photosynthesis
being in general use, simulated photosynthesis varies significantly between LSMs. This variation is attributed to
several factors, including 1) differences in prescribed Rubisco kinetic constants and their temperature responses
(Rogers et al., 2017), 2) structural differences, namely the method used to determine the transition point between
the limiting rates of photosynthesis which has a disproportionate impact on estimates (Huntingford and Oliver,
2021; Walker et al., 2021) , and 3) the sensitivity of photosynthesis to temperature, in terms of the under-
representation of parameters from different biomes to describe the short-term instantaneous response of
photosynthesis to temperature (Rogers et al., 2017). In particular, these differences imply that parameter values
derived calibrating the Collatz model against data will differ to those derived using Farquhar against the same set
of measurements. Parameter values are not transferable between models, hence such differences will lead to
inconsistencies and projection errors if parameters are fitted to data, but then applied within the alternative model.
Building in the capacity of an LSM to run with either photosynthesis scheme greatly enhances flexibility in
modelling. Importantly, this flexibility allows for consistency between parameters used by empiricists to derive
leaf level photosynthetic parameters from observations, and those used in large scale modelling. Additionally, our
re-parameterisation of the photosynthetic capacity and temperature sensitivity parameters are based on recent
global datasets that are more extensive, including species from a range of different biomes, further enhancing the
capacity for global modelling applications. Our second rationale is that the Farquhar photosynthesis scheme is
required as the underlying model to implement the Kattge and Knorr (2007) thermal acclimation scheme.
Leaf level $g_s$ response to water vapour is commonly represented in LSMs empirically (Jarvis et al., 1976), or with
a semi-empirical model (Ball et al., 1987; Damour et al., 2010; Leuning, 1995). Values of $g_s$ are subsequently
scaled yielding an estimate of canopy conductance for vegetation in different ecosystems. De Kauwe et al. (2013)
showed that 10 of the 11 ecosystem models studied in their inter-comparison used a form of the "Ball–Berry–
Leuning" approximation. This model form links $g_s$ to changes in environmental conditions, and directly to
photosynthetic rate. However, there is increasing interest in using models based on optimisation theory (Franks et
al., 2017; Franks et al., 2018), using evidence that stomata may behave to maximise $CO_2$ gain whilst minimising
water loss. The major advantage of optimality theory is that the optimisation criterion will apply under any
environmental conditions, past or future. Hence the derived equations can replace uncertain mechanistic
formulations and may also have more predictive capability corresponding to future climate regimes. JULES
traditionally uses the empirically-based Jacobs (1994) $g_s$ scheme ("Jacobs"), and in this study we compare the
behaviour of this scheme against the Medlyn et al. (2011) $g_s$ scheme ("Medlyn") which is based on optimisation
theory. The Medlyn $g_s$ model has been previously implemented in JULES by Oliver et al. (2018). However, in
this study, we advance on that previous work by calibrating for the increased number of plant functional types
now in JULES (nine PFTs, as opposed to five in the original study), and we parameterise using data from a global
synthesis of experimental observations.
There is increasing evidence that the short-term vegetation temperature responses are themselves sensitive to
temperatures experienced over longer time-scales (days to weeks to seasons) and in particular, have the capability
to acclimate to growth temperature ($T_{growth}$) (Kattge and Knorr, 2007). Observational evidence of thermal
acclimation of photosynthesis has been widely reported, primarily for temperate and boreal ecosystems (Atkin et
al., 2006; Gunderson et al., 2000; Gunderson et al., 2010; Hikosaka et al., 2007; Way and Yamori, 2014; Yamori
et al., 2014). The effect is defined as the fast temporal adjustment of the temperature response of photosynthesis
driven by a change in $T_{growth}$. Thermal acclimation of photosynthesis typically results in a shift in the optimum
temperature ($T_{opt}$) for photosynthesis towards the new growth temperature, which can result in an increase or
maintenance of the photosynthetic rate respective to $T_{growth}$ (Yamori et al., 2014). In this study, we implement
thermal acclimation of photosynthetic capacity in JULES using the scheme from Kattge and Knorr (2007). The
scheme attributes all changes in the photosynthetic response to changing $T_{growth}$, without specifically separating
adaptation from acclimation processes. Of those LSMs that do account for thermal acclimation of photosynthesis
(e.g. TEM, CLM4.5, LM3, JULES) (Chen and Zhuang, 2013; Lombardozzi et al., 2015; Mercado et al., 2018;
Smith et al., 2016), all similarly use this numerical algorithm from Kattge and Knorr (2007). Mercado et al. (2018)
investigated the impacts of thermal acclimation on the future land carbon sink using an implementation of the
Kattge and Knorr (2007) in JULES, although using a simple big leaf scaling approach. In this study we apply the
thermal acclimation scheme in the updated JULES model (i.e. newly parameterised Farquhar scheme, running
with a multi-layer canopy and nine PFTs) and updated with the Medlyn $g_s$ scheme and related parameters.
This paper therefore brings together these three key recent developments of the JULES plant physiology routines,
(1) implementation of the Farquhar photosynthesis scheme, (2) the optimisation-based Medlyn model of stomatal
opening, and (3) thermal acclimation of photosynthesis, along with updated parameters and an evaluation of model
behaviour. We make incremental additions of the different processes to the JULES model in a set of factorial
simulations and run the model with current day (1979 to 2013) near-surface meteorological forcing and $CO_2$
levels. First, we present the different factorial simulations in the context of a thorough evaluation of simulated
contemporary carbon and energy fluxes. Such evaluation includes comparison against individual eddy covariance
sites, and at spatial scales up to the global scale against satellite products. Timescales analysed are both seasonal
and annual. Secondly, we apply the new model configurations within a past-to-future climate change simulation
based on a high-end emissions scenario (SSP5 RCP8.5). We use output from HadGEM3-GC3.1 spanning years
1960 to 2050 to explore sensitivity of global vegetation to future climate change. This choice of scenario is to
allow eventual comparison between these offline simulations and the equivalent in the coupled global climate
model to investigate land-atmosphere feedbacks resulting from these changes to the plant physiology routines.
This is currently work being undertaken. This updated version of the JULES model is now available in official
JULES releases for use by the community (see data availability). It is therefore also readily available for full
coupling into the UK community ESM (UKESM), a process that is just starting.

**2. Model description**
**2.1 JULES land surface model**
Our modelling framework is JULES (https://jules.jchmr.org), the land surface component of the Hadley Centre
climate models, which includes the new UK community Earth System Model (UKESM1) (Sellar et al., 2019).
JULES can be run offline, as in this study, forced with observed meteorology, at different spatial scales (from a
single location to global). A full description of JULES is provided in Best et al. (2011), Clark et al. (2011) and
Harper et al. (2016). Of particular relevance for this study is the plant physiological representation in JULES.
JULES uses a leaf-level coupled model of photosynthesis and $g_s$ (Cox et al., 1998) based on Collatz et al. (1991)
and Collatz et al. (1992) (for $C_3$ and $C_4$ plants) and Jacobs (1994) respectively. Photosynthesis and $g_s$ are modelled
to respond to changes in environmental drivers of temperature, humidity deficit, light, $CO_2$ concentration and
water availability. Soil moisture content is modelled using a dimensionless soil water stress factor which is related
to the mean soil water concentration in the root zone, and the soil water contents at the critical and wilting point
(Best et al., 2011). The critical and wilting point soil moisture concentrations vary by soil type in these simulations.
In this study, JULES uses a multilayer canopy radiation interception and photosynthesis scheme (i.e. 10 layers)
that accounts for vertical variation of incoming direct and diffuse radiation, sun fleck penetration through the
canopy, change in photosynthetic capacity with depth into the canopy, inhibition of leaf respiration in the light
and differentiates calculation of sunlit and shaded photosynthesis at each layer (Clark et al., 2011; Mercado et al.,
2009). The implementation of a multilayer canopy for light interception in JULES was shown to improve modelled
canopy scale photosynthetic fluxes at eddy covariance sites compared to the 'big leaf approach' (Blyth et al.,
2011; Jogireddy et al., 2006; Mercado et al., 2007). Specifically, the multi-layer approach better captured the light
response and diurnal cycles of canopy photosynthesis. While light inhibition of leaf respiration and changing
photosynthetic capacity with canopy depth are supported by observations (Atkin et al., 2000; Atkin et al., 1998;
Meir et al., 2002). Sunfleck penetration through the canopy and the differential effects of direct and diffuse beam
radiation on modelled carbon and water exchange in JULES were studied by Mercado et al. (2009). This enabled
JULES to reproduce the different light-response curves of GPP under diffuse and direct radiation conditions at
both a broadleaf and needleleaf temperate forest.
**2.2 Physiology Developments**
**2.2.1 Farquhar photosynthesis for $C_3$ plants and parameterisation**
We implement the Farquhar photosynthesis scheme (Farquhar et al., 1980) to describe the leaf-level biochemistry
of photosynthesis for $C_3$ vegetation following the approach of Mercado et al. (2018). Here the leaf-level
photosynthesis is calculated as the minimum (note no smoothing) of two potentially limiting rates (Equation 1a).
These two rates are i) Rubisco-limited photosynthesis (Equation 2) and ii) light-limited photosynthesis with a
dependence on the incident photosynthetically active photon flux density and the potential electron transport rate
(Equations 3 and 4). Note, as in the original Farquhar formulation, we do not include a TPU-limited (triose
phosphate utilisation) rate. Further, recent empirical studies suggest that TPU limitation rarely limits
photosynthesis under present-day $CO_2$ concentrations and is also unlikely to limit photosynthesis at elevated $CO_2$
(Kumarathunge et al., 2019a). This, and the current uncertainty in the formulation of TPU limitation of
photosynthesis led Rogers et al. (2021) to conclude it is an unnecessary complication in LSMs. Hence:
$A_p = min\{A_v, A_j\} - R_d$                                                                   (1a)
$A_n = A_p \beta$                                                                           (1b)
where $A_p$ is the net potential (i.e. unstressed) leaf photosynthetic carbon uptake (mol m$^2$ s$^{-1}$), $R_d$ is the rate of leaf
respiration in the dark (mol m$^2$ s$^{-1}$), $A_n$ is the net photosynthetic rate (mol m$^2$ s$^{-1}$) which accounts for the impact of
soil moisture stress on photosynthetic rate by multiplying $A_p$ by the soil water stress factor $\beta$. Rubisco-limited
photosynthesis ($A_v$, mol m$^2$ s$^{-1}$) is calculated as in Equation 2. The maximum rate of carboxylation of Rubisco is
determined by $V_{cmax}$ (mol m$^2$ s$^{-1}$), $c_i$ and $o_a$ are the intercellular concentrations of $CO_2$ and $O_2$ (both Pa), $K_c$ and $K_o$
(both units of Pa) are the Michaelis Menten coefficients for Rubisco carboxylation and oxygenation respectively,
and $\Gamma$ (Pa) is the $CO_2$ compensation point in the absence of mitochondrial respiration.

$A_v = \dfrac{V_{cmax}\,(c_i - \Gamma)}{\left[c_i + K_c\left(1 + \frac{o_a}{K_o}\right)\right]}$                                                       (2)
The light-limited rate of photosynthesis ($A_j$, mol m$^2$ s$^{-1}$) (Equation 3) is a function of the rate of electron transport
$J$ (mol m$^2$ s$^{-1}$) which is represented in Equation 4. $J$ depends on the incident photosynthetically active photon flux
density $Q$ (mol quanta m$^2$ s$^{-1}$), the potential rate of electron transport $J_{max}$ (mol m$^2$ s$^{-1}$), the apparent quantum yield
of electron transport $\alpha$ (mol electrons mol$^{-1}$ photon) fixed at 0.3 (mol electrons mol$^{-1}$ photon) following Medlyn
et al. (2002), and $\theta$ a non-rectangular hyperbola smoothing parameter which takes a value of 0.9 (unitless)
following Medlyn et al. (2002). The factor of four used in the Farquhar model in Equation 3 accounts for four
electrons being required per carboxylation/oxygenation reaction.

$A_j = \left(\dfrac{J}{4}\right)\dfrac{(c_i - \Gamma)}{(c_i + 2\Gamma)}$                                                              (3)

$\theta J^2 - (\alpha Q + J_{max})J + \alpha Q J_{max} = 0$                                         (4)

JULES currently uses $Q_{10}$ functions in the Collatz scheme to describe the temperature dependency of $V_{cmax}$, $K_c$,
$K_o$, and $\Gamma$ (see Notes S1). In our implementation of the Farquhar scheme, temperature sensitivities for the $K_c$, $K_o$,
and $\Gamma$ are taken from Bernacchi et al. (2001) as described in Medlyn et al. (2002). These are the same temperature

sensitivities used by experimentalist to derive estimates of photosynthetic capacity parameters (Rogers et al., 2017). Of particular importance to our analysis here are the temperature responses of $V_{cmax}$ and $J_{max}$. Equation 5 describes the temperature response of both parameters:

$$k_T = k_{25}\, exp\left[H_a\, \frac{(T_l - T_{ref})}{T_{ref}\, R\, T_l}\right] \frac{1 + exp\left[\frac{T_{ref}\, \Delta S - H_d}{T_{ref}\, R}\right]}{1 + exp\left[\frac{T_l\, \Delta S - H_d}{T_l\, R}\right]} \tag{5}$$

Here, $k_T$ ($\mu mol\ m^2\ s^{-1}$) is either $V_{cmax}$ or $J_{max}$ at leaf temperature $T_l$ (K), $k_{25}$ ($\mu mol\ m^2\ s^{-1}$) is the rate of $V_{cmax}$ or $J_{max}$ at the reference temperature $T_{ref}$ of 25 ºC (298.15 K), $R$ is the universal gas constant (8.314 J mol$^{-1}$ K$^{-1}$), $H_a$ and $H_d$ (J mol$^{-1}$) are the activation and deactivation energies respectively, and $\Delta S$ (J mol$^{-1}$ K$^{-1}$) is an entropy term (see Table 1 for PFT-specific parameter values). Broadly, $H_a$ describes the rate of exponential increase of the function below the optimum temperature ($T_{opt}$), and $H_d$ describes the rate of decrease above the $T_{opt}$. $\Delta S$ and $T_{opt}$ are related by Equation 6, which is used to calculate the $T_{opt}$ of $V_{cmax}$ and $J_{max}$ (Table 1):

$$T_{opt} = \frac{H_d}{\Delta S - R\, ln\left[\frac{H_a}{H_d - H_a}\right]} \tag{6}$$

**Table 1.** PFT-specific parameters for the required temperature dependency of $V_{cmax}$ and $J_{max}$ in the Collatz and Farquhar photosynthesis schemes. PFT codes (left column) are BET-tr: Broadleaf evergreen tropical tree, BET-te: Broadleaf evergreen temperate tree, BDT: Broadleaf deciduous tree, NET: Needle leaf evergreen tree, NDT: Needle leaf deciduous tree, C$_3$: C$_3$ grass, C$_4$: C$_4$ grass, ESH: Evergreen shrub, DSH: Deciduous shrub.

| | Collatz | | | Farquhar | | | | | | |
|---|---|---|---|---|---|---|---|---|---|---|
| | $T_{upp}$ | $T_{low}$ | $Topt_{vcmax}$ | $Ha_{vcmax}$ | $Ha_{jmax}$ | $\Delta s_{vcmax}$ (J mol$^{-1}$ K$^{-1}$) | $\Delta s_{jmax}$ (J mol$^{-1}$ K$^{-1}$) | $Topt_{vcmax}$ | $Topt_{jmax}$ | $Hd_{vcmax}$ or $Hd_{jmax}$ |
| | (ºC) | (ºC) | (ºC) | (J mol$^{-1}$) | (J mol$^{-1}$) | | | (ºC) | (ºC) | (J mol$^{-1}$) |
| BET-tr | 43 | 13 | 39.00 | 86900 | 64000 | 631 | 635 | 42.71 | 38.73 | 200000 |
| BET-te | 43 | 13 | 39.00 | 59600 | 35900 | 634 | 632 | 38.80 | 37.10 | 200000 |
| BDT | 43 | 5 | 39.00 | 49300 | 38800 | 658 | 663 | 26.57 | 23.22 | 200000 |
| NET | 37 | 5 | 33.00 | 63100 | 36400 | 642 | 643 | 35.28 | 31.96 | 200000 |
| NDT | 36 | -5 | 34.00 | 49300 | 38800 | 658 | 663 | 26.57 | 23.22 | 200000 |
| C$_3$ | 32 | 10 | 28.00 | 97200 | 112000 | 660 | 663 | 28.00 | 28.00 | 199000 |
| C$_4$ | 45 | 13 | 41.00 | - | - | - | - | - | - | - |
| ESH | 36 | 10 | 32.00 | 59600 | 35900 | 634 | 632 | 38.80 | 37.10 | 200000 |
| DSH | 36 | 0 | 32.00 | 49300 | 38800 | 658 | 663 | 26.57 | 23.22 | 200000 |

To find new estimates for $V_{cmax}$ and the $J_{max}:V_{cmax}$ ratio at $T_{ref}$ of 25°C for use with the Farquhar model for the 9
PFT's in JULES we used the global dataset from Walker et al. (2014) which includes data from 356 species. For
$V_{cmax}$ and $J_{max}$, Walker et al. (2014) re-analysed the data to remove the variation in these two parameters across
studies caused by different parametric assumptions used in their derivation from $A$-$C_i$ curves (e.g. using a common
set of kinetic parameters, and reporting values at 25°C). We calculated the mean $V_{cmax}$ and $J_{max}$ across studies
conducted at ambient $CO_2$ concentration for each of the JULES PFTs (Table 2). To parameterise the deciduous
needleleaf tree (NDT) PFT, we use the values for the evergreen needleleaf tree (NET) PFT because the data for
NDT was from a single study on one juvenile (3 years old) species. An exception was the tropical broadleaf
evergreen tree (BET-tr) PFT, where we use $V_{cmax}$ and $J_{max}$ from the dataset collated in the more recent compilation
by Kumarathunge et al. (2019b), as this study includes many more tropical tree species than any previous meta-
analysis.
Parameter values for the temperature response functions for $V_{cmax}$ and $J_{max}$ (Equation 5) in the Farquhar scheme
were taken from a global dataset of photosynthetic $CO_2$ response curves, which entrained data from 141 $C_3$
species, ranging from the tropical rainforest to Arctic tundra (Kumarathunge et al., 2019b). The study provides
parameter values for tree PFT's that match those in JULES, e.g. tropical broadleaf evergreen trees (BET-tr PFT
in JULES), temperate broadleaf evergreen trees (BET-te), broadleaf deciduous trees (BDT) and needleleaf
evergreen trees (NET). For the remaining JULES PFTs, BDT values are used for NDT and deciduous shrubs
(DSH), and BET-te values are used for evergreen shrubs (ESH). Kumarathunge et al. (2019b) do not include data
for $C_3$ grasses, therefore to parameterise the temperature dependency of $V_{cmax}$ and $J_{max}$ for this PFT, we fitted both
to the existing $V_{cmax}$ temperature response function in the Collatz scheme for $C_3$ grasses because of a scarcity of
data in the literature. Fig. S1 shows the temperature dependency of $V_{cmax}$, $J_{max}$ and gross photosynthesis for Collatz
and Farquhar using the PFT-specific parameters in Table 1 and Table 2.

**Table 2.** PFT-specific parameters for the Collatz and Farquhar photosynthesis schemes.

| | Collatz | | Farquhar | | | |
|---|---|---|---|---|---|---|
| | $V_{cmax25}$ (umol m$^2$s$^{-1}$) | $\alpha$ (intrinsic) (mol $CO_2$ mol$^{-1}$ PAR) | $V_{cmax25}$ (umol m$^2$s$^{-1}$) | $J_{max25}$ (umol m$^2$s$^{-1}$) | $J_{max}:V_{cmax}$ | $\alpha$ (apparent) (mol electrons mol$^{-1}$ photon) |
| BET-tr | 41.16 | 0.08 | 39.50 | 63.20 | 1.60 | 0.30 |
| BET-te | 61.28 | 0.06 | 68.95 | 112.59 | 1.63 | 0.30 |
| BDT | 57.25 | 0.08 | 55.24 | 98.30 | 1.78 | 0.30 |
| NET | 53.55 | 0.08 | 50.80 | 75.14 | 1.48 | 0.30 |
| NDT | 50.83 | 0.10 | 50.80 | 75.14 | 1.48 | 0.30 |
| $C_3$ | 51.09 | 0.06 | 43.83 | 108.07 | 2.47 | 0.30 |
| $C_4$ | 31.71 | 0.04 | - | - | - | - |
| ESH | 62.41 | 0.06 | 68.96 | 112.59 | 1.63 | 0.30 |
| DSH | 50.40 | 0.08 | 55.24 | 98.30 | 1.78 | 0.30 |


### 2.2.2 Medlyn model of $g_s$ and parameterisation

In JULES, $g_s$ (m s$^{-1}$) is represented in Equation 7.

$$g_s = 1.6RT_l \frac{A_n}{c_a - c_i} \qquad (7)$$

where the factor 1.6 accounts for $g_s$ being the conductance for water vapour rather than $CO_2$, $R$ is the universal gas constant (J mol$^{-1}$ K$^{-1}$), $T_l$ is the leaf surface temperature (K), $c_a$ and $c_i$ (both Pa) are the leaf surface and internal $CO_2$ partial pressures respectively, and $A_n$ is the net photosynthetic rate. Here, $c_i$ is unknown and is calculated in JULES using the Jacobs scheme as in Equation 8, and relates the ratio of ambient ($c_a$) to leaf intercellular ($c_i$) partial pressure of $CO_2$ ($c_i/c_a$), to leaf humidity deficit:

$$c_i = (c_a - \Gamma)f_0 \left(1 - \frac{d_q}{dq_{crit}}\right) + \Gamma \qquad (8)$$

where $\Gamma$ (Pa) is the $CO_2$ photorespiration compensation point, $d_q$ is the humidity deficit at the leaf surface (kg kg$^{-1}$), and $dq_{crit}$ (kg kg$^{-1}$) and $f_0$ are PFT specific parameters representing the critical humidity deficit at the leaf surface and the leaf internal to atmospheric $CO_2$ ratio ($c_i/c_a$) at the leaf specific humidity deficit (Best et al., 2011). To implement the Medlyn model, Equation 9 is used to calculate $c_i$, retaining Equation 7 to calculate $g_s$. In Equation 9, $g_1$ (kPa$^{0.5}$) is a PFT-specific model parameter and $d_q$ is expressed in kPa. The Medlyn scheme is based on optimisation theory, and so assumes that stomatal aperture is regulated to maximize carbon gain while simultaneously minimising water loss:

$$c_i = c_a \left(\frac{g_1}{g_1 + \sqrt{d_q}}\right) \qquad (9)$$

PFT-specific values of the $g_1$ parameter were derived for the nine JULES PFTs from the global data base of Lin et al. (2015) (Table 3). The $g_1$ parameter represents the sensitivity of $g_s$ to the assimilation rate, i.e. plant water use efficiency, and was derived as in Lin et al. (2015), by fitting the Medlyn et al. (2011) model to observations of $g_s$, photosynthesis, and VPD, assuming an intercept of zero. A non-linear mixed-effects model was used to estimate the model slope coefficient, $g1$, for each PFT, where individual species were assumed to be the random effect to account for the differences in the $g1$ slope among species within the same group, following Lin et al. (2015).

**Table 3.** PFT-specific parameters required for the Jacobs and Medlyn $g_s$ schemes.

| | Jacobs $f_o$ | Jacobs $dq_{crit}$ (kg kg$^{-1}$) | Medlyn $g_1$ (kPa$^{0.5}$) |
|---|---|---|---|
| BET-tr | 0.875 | 0.090 | 5.31 |
| BET-te | 0.892 | 0.090 | 3.37 |
| BDT | 0.875 | 0.090 | 4.45 |
| NET | 0.875 | 0.060 | 2.35 |
| NDT | 0.936 | 0.041 | 2.35 |
| C$_3$ | 0.931 | 0.051 | 5.25 |
| C$_4$ | 0.800 | 0.075 | 1.62 |
| ESH | 0.950 | 0.037 | 3.29 |
| DSH | 0.950 | 0.030 | 5.47 |


### 335    2.2.3 Thermal acclimation of photosynthetic capacity

The Kattge and Knorr (2007) acclimation algorithm ("AcKK") is based on the parameters of the Farquhar
photosynthesis scheme, hence acclimation is implemented in the Farquhar model. The AcKK algorithm uses
empirical relationships to describe the response of $V_{cmax}$, $J_{max}$, and the $J_{max}{:}V_{cmax}$ ratio to changes in $T_{growth}$ (defined
in AcKK as the average temperature (day and night) of the previous 30 days), and importantly it represents
combined acclimation and adaptation processes. Kattge and Knorr (2007) found that $\Delta S_v$, $\Delta S_j$, and the $J_{max}{:}V_{cmax}$
ratio decrease linearly with increasing $T_{growth}$ following Equation 10. This means according to these relationships,
the optimum temperatures ($T_{opt}$) of $V_{cmax}$ and $J_{max}$ ($T_{optv}$ and $T_{optj}$) increase by 0.44°C and 0.33°C per degree increase
in $T_{growth}$ respectively, and the $J_{max}{:}V_{cmax}$ ratio at 25°C decreases by 0.035°C per degree increase in $T_{growth}$.
$$x_i = a_i + b_i\, T_{growth} \tag{10}$$
The $x$ is either $\Delta S_v$, $\Delta S_j$ or the $J_{max}{:}V_{cmax}$ ratio, and the sub-index $i$ refers to the parameter values ($a$ and $b$ shown in
Table 4) for $V_{cmax}$, $J_{max}$ or the $J_{max}{:}V_{cmax}$ ratio. $T_{growth}$ is the growth temperature (calculated online as the mean
temperature of the previous 30 days).

**Table 4.** Parameter values derived by Kattge & Knorr (2007) and used in this study in Equation 10 to model
thermal acclimation of photosynthesis using the AcKK scheme.

| | Acclimation | |
|---|---|---|
| | $a$ | $b$ |
| $\Delta S_j$ | 659.7 | -0.75 |
| $\Delta S_v$ | 668.39 | -1.07 |
| $J_{max}{:}V_{cmax}$ | 2.59 | -0.035 |

**3. Model evaluation and application**

**3.1 Site level simulations**

JULES was applied using four model configurations (Table 5) with observed meteorology, and evaluated against data from 17 eddy covariance sites (Table S1, Fig. S2). This collection of eddy covariance measurements represents a range of climates and land cover types (Table S1, Fig. S2). In all simulations the vegetation cover was prescribed, removing any biases that the modelled competition may introduce through self-diagnosis of PFT extents. Prescribed leaf area index (*LAI*) was used where site data was available, otherwise the JULES phenology scheme was switched on allowing the model to evolve the *LAI*. Model output was evaluated against fluxes of gross primary productivity (GPP) and evaporative fraction (EF). We used EF rather than latent heat flux to minimise issues with incomplete closure of the energy balance (that can typically range from 5 to 30 % at some eddy covariance sites, Liu et al. (2006)). For analysis we used daytime values only (i.e. where the shortwave radiation was > 10 W m$^2$) from days with no missing data, and compare mean seasonal diurnal cycles of modelled GPP and EF against the observed fluxes. The mean seasonal cycle calculated over the entire measurement period is used in order to assess the mean model behaviour.

We evaluate the site-level simulations with RMSE (root mean square error) for the seasonal diurnal cycle of simulated (daytime) fluxes (GPP and EF). For each site, the time period of the simulation and therefore evaluation period is stated in Table S1. We summarise the changes in RMSE using the relative improvement for each model configuration (*i*) compared to the current standard JULES configuration of Collatz with Jacobs (Clz.Jac). The statistic is calculated so that positive values show an improvement compared to Clz.Jac and therefore a better comparison to the observations:

$$RMSE_{rel_i} = \frac{RMSE_{Clz.Jac} - RMSE_i}{RMSE_{Clz.Jac}} \tag{11}$$

**Table 5.** Description of the four model experiments performed both at site level and globally, with the JULES land surface model.

| Model simulation | Description | Photosynthesis scheme | Stomatal closure | Temperature dependency of photosynthesis | $T_{growth}$ |
|---|---|---|---|---|---|
| **Clz.Jac** | The original photosynthesis and stomatal conductance ($g_s$) schemes used in JULES. | Collatz *et al.*, (1991) | Jacobs (1994) | $Q_{10}$ function for $K_c$, $K_o$ $\Gamma$ and $V_{cmax}$ (PFT specific). $T_{opt}$ varies by PFT but is fixed spatially and temporally. | NA |
| **Fq.Jac** | The Farquhar photosynthesis scheme is implemented with updated $V_{cmax}$ and $J_{max}$ values, and updated parameters for the temperature response of photosynthesis ($\Delta S$ and $H_a$ for $V_{cmax}$ and $J_{max}$) with original $g_s$ scheme used in JULES. | Farquhar *et al.*, (1980) | Jacobs (1994) | Arrhenius function for $K_c$, $K_o$ $\Gamma$, $V_{cmax}$ and $J_{max}$ (latter two both PFT specific). $T_{opt}$ varies by PFT but is fixed spatially and temporally. | NA |
| **Fq.Med** | The Medlyn stomatal closure is implemented with the parameter $g_1$ that varies by PFT with Farquhar photosynthesis model implementation. | Farquhar *et al.*, (1980) | Medlyn *et al.*, (2011) | Arrhenius function for $K_c$, $K_o$ $\Gamma$, $V_{cmax}$ and $J_{max}$ (latter two both PFT specific). $T_{opt}$ varies by PFT but is fixed spatially and temporally. | NA |
| **AcKK.Med** | Thermal acclimation of photosynthetic capacity accounted for. Implemented within the Farquhar model coupled to the Medlyn $g_s$ model. | Farquhar *et al.*, (1980) | Medlyn *et al.*, (2011) | Arrhenius function for $K_c$, $K_o$ $\Gamma$. Thermal acclimation of photosynthetic capacity implemented following Kattge & Knorr (2007). Parameters describing the temperature sensitivity of photosynthesis ($\Delta S$ for $V_{cmax}$ and $J_{max}$, and the $J_{max}$:$V_{cmax}$) allowed to acclimate to the temperature of the growth environment ($T_{growth}$). $T_{opt}$ adjusts to changes in $T_{growth}$ so varies spatially and temporally. | Yes |

**3.2 Global scale simulations**

Four JULES simulations were performed globally for the period 1979-2013 as outlined in Table 5. These global present-day simulations were run at 0.5º x 0.5º spatial resolution. The WFDEI meteorological dataset was used to drive the model (Weedon et al., 2014). This has a three hour temporal resolution that JULES interpolated down to an hourly model time step. To focus on the direct effects of the model changes on GPP and surface energy fluxes, the land surface properties of the model were prescribed. We use a static map of land cover (in terms of

different PFT extents) derived from the European Space Agency's Land Cover Climate Change Initiative (ESA LC_CCI) global vegetation distribution version 1.6 for the 2010 epoch (Poulter et al., 2015) (Fig. S3) following that used in Harper et al. (2016). Seasonally varying *LAI* were derived from the Global LAnd Surface Satellite (GLASS) dataset (Xiao et al., 2016). Prescribed parameters were used for the hydraulic and thermal properties of the soil from a modified version of the H1 lookup-table from Zhang and Schaap (2017) that depends upon the soil textural type from SoilGrids (Hengl et al., 2014).We also prescribe transient atmospheric $CO_2$ concentrations based on annual mean observations from Mauna Loa (Tans and Keeling, 2014). A spin-up of 80 years was performed (re-cycling through the period 1979 to 1999), which is sufficient to equilibrate soil temperature and soil moisture.

The global offline present-day simulations were compared against the global evaluation products, and for both model output and observations we calculate seasonal means over the period 2002 to 2012. We used the global FluxCom product to evaluate modelled GPP, LE, H and ET (Jung et al., 2020; Tramontana et al., 2016). We compare our simulations against the FluxCom ensemble product (RS+MET) driven with the same forcing (WFDEI), as is recommended by Jung et al. (2019) to minimise deviations due to different climate input data. To convert LE to ET we assume a constant latent heat of vaporization of 2.5 MJ mm$^{-1}$. We also use the model derived product from GLEAM-v3.3a to evaluate ET, and additionally use the MODIS GPP product (Zhao et al., 2005; Zhao and Running, 2010; Zhao et al., 2006) to evaluate simulated global GPP.

Global future climate simulations were performed forced with meteorological output (1960 to 2050) from the HadGEM3-GC3.1 model atmosphere-only simulations at 3 hour temporal resolution and N512 spatial resolution (Roberts et al., 2019; Williams et al., 2018). These projections follow the CMIP6 HighResMIP protocol (Haarsma et al., 2016). This choice of forcing to drive JULES is to allow comparison of the offline runs performed in this study with the equivalent simulations currently being undertaken in the coupled HadGEM3-GC3.1 model to explore land-atmosphere feedbacks arising from changes implemented to the plant physiology routines in this work. The factorial set of offline simulations in this work provide a systematic sensitivity study that is less computationally expensive with which to help understand behaviour seen in the coupled model. The output at N512 was re-gridded to 0.5º x 0.5º using conservative interpolation which ensures the physical conservation of each variable. Fig. S4 shows the mean temperature and precipitation change by region over the study period, and the atmospheric $CO_2$ concentration. Atmospheric $CO_2$ concentrations were prescribed based on observations up to 2014 as described in historical CMIP6 simulations (Eyring et al., 2016). From 2015 onwards, atmospheric $CO_2$ concentrations were based on a high-end emission scenario of the Shared Socioeconomic Pathways (SSP5) with the Representative Concentration Pathway 8.5 (RCP8.5) (Haarsma et al., 2016). As for the current-day simulations, *LAI*, land cover and soil properties were prescribed using the same datasets. A spin-up period of 80 years (re-cycling through the period 1960 to 1980) was again used to equilibrate soil temperature and soil moisture.

We analyse the future global simulations using the 'difference of difference' approach. This method explicitly targets the change in the variable of interest over the study period resulting from the change in process alone, and negates differences that may arise from different initial starting points of each simulation (different initial conditions):

$$Effect = (\bar{X}_{2050} - \bar{X}_{1980}) - (\bar{Y}_{2050} - \bar{Y}_{1980}) \tag{12}$$
where $X$ represents the simulation with the process of interest and $Y$ represents the simulation with the alternative
representation, and 2050 and 1980 represent the end and start of the simulation analysis period respectively
(calculated as the mean over 2040 to 2050, and 1980 to 1990 respectively). For example, to look at the impact of
changing photosynthesis schemes, $X$ = Fq.Jac and $Y$ = Clz.Jac. In this case, both configurations are using the
Jacobs $g_s$ scheme, only the photosynthesis scheme changes from Collatz to Farquhar. The impact of changing $g_s$
scheme is assessed where $X$ = Fq.Med and $Y$ = Fq.Jac. The impact of thermal acclimation is assessed where $X$ =
AcKK.Med and $Y$ = Fq.Med, here both simulations use the Farquhar photosynthesis scheme and the Medlyn $g_s$
scheme, but $X$ has the addition of thermal acclimation of photosynthesis.

**4. Results**

**4.1 Site level evaluation**

Results from the FLUXNET sites comparing the mean seasonal diurnal cycles of GPP and EF against observed
fluxes are summarised in Fig. 1, where reds and yellows indicate reduced RMSE relative to the 'standard' JULES
configuration of Collatz with Jacobs (Clz.Jac), and therefore closer agreement to site level FLUXNET
observations. Results are variable by site and season (Fig. 1, Fig. S5 and Fig. S6), some of which will be due to
other site-specific characteristics that are not simulated well by the model, such as *LAI* for those sites that rely on
model derived estimates. On the other hand, soil properties are prescribed by parameters that describe the thermal
and hydraulic characteristics of the soil, uncertainties in these parameterisations have consequences for the
simulated soil moisture content at each site, for example, which impacts simulated carbon and water fluxes. We
first consider results for the five tropical sites. Results are mixed for the simulated seasonal diurnal cycle of GPP
at the tropical (EBF / BET-tr) sites, GPP is improved (reduced) with the new JULES model configurations at three
out of the five tropical sites in March-April-May (MAM; Fig. 1a, Fig. S5), with thermal acclimation leading to
the greatest improvements. However in June-July-August (JJA; Fig. 1b, Fig. S5), this improvement is only found
at two of the tropical sites. At the EBF sites, implementing the Farquhar photosynthesis model means $V_{cmax}$ is
lower (BET-tr, Table 2), and this in addition to the change in temperature sensitivity (Table 1; Fig. S1a-c), and
model structural changes from Collatz to Farquhar results in lower simulated GPP compared to Collatz. Thermal
acclimation allows further adjustments of the $T_{opt}v$, $T_{opt}j$ and the $J_{max}:V_{cmax}$ ratio which results in lower simulated
photosynthesis and therefore GPP compared to Farquhar (Fig. S5). The change from Jacobs $g_s$ model to Medlyn
has minimal impact on simulated GPP for the tropical tree PFT because in both schemes the modelled $c_i$ has a
similar sensitivity to humidity deficit at the leaf surface, with the exception at very low humidity deficit (Fig. S7;
Fig. S5). The simulated seasonal diurnal cycle of EF is improved (reduced) at four out of the five tropical sites in
both MAM and JJA, again with some of the largest improvements seen with thermal acclimation (Fig. 1c & 1d,
Fig. S6).
At the C₃ grassland sites (GRA), improved simulated GPP (higher GPP) is seen across all sites in JJA with the
Medlyn $g_s$ scheme and thermal acclimation (Fig. 1b, Fig. S5). This is matched by improvements in simulated EF
(higher EF) across all grassland sites in both seasons, with the exception of US_var in JJA (Fig. 1c & 1d; Fig. S6).
The change from Collatz to Farquhar at the GRA sites means a lower $V_{cmax}$ is used (C3, Table 2) although the
temperature sensitivity is similar (Table 1, Fig. S1p, q), this results in lower GPP simulated by Farquhar compared
to Collatz which compares worse to the observations (GPP and EF, Fig. 1, Fig. S5). In contrast to using Farquhar
with the Jacobs $g_s$ scheme, using Farquhar with the Medlyn scheme improves simulated GPP and EF, both are
increased because for the C3 grass PFT as the humidity deficit at the leaf surface increases $c_i$ simulated by Medlyn
is less sensitive compared to Jacobs (Fig. S7; Fig. S5), leading to higher $c_i$, higher net canopy photosynthesis and
GPP, and higher transpiration and LE. These results suggests the Medlyn scheme has a large impact on simulated
carbon and water fluxes for the C3 grass PFT in the JULES model. In JJA, the adjustment of the temperature
sensitivity of photosynthesis to the $T_{growth}$ by the thermal acclimation scheme tends to increase GPP compared to
Farquhar with no acclimation, and this compares better to the observations (Fig. 1, Fig. S5).
At the broadleaf deciduous tree sites (BDT) simulated GPP is improved with all JULES model configurations in
MAM (higher GPP) at three out of the four sites (Fig. 1a). However in JJA improvements are mainly seen with
thermal acclimation (lower GPP compared to Fq.Med, Fig. 1b). Medlyn $g_s$ performs worse at all sites in JJA
suggesting either the model formulation or parameters are not suitable to correctly capture stomatal behaviour in
this season for this PFT (Fig. 1b, Fig. S5). Compared to Collatz, the Farquhar model for the BDT PFT uses a
lower $V_{cmax}$ (Table 2) and has a considerably lower $T_{opt}v_{cmax}$ (Table 1; Fig. S1h), which means that at leaf
temperatures below ~22°C, photosynthesis is higher with the Farquhar model, and above this photosynthesis is
lower than Collatz (Fig. S1g). Consequently, warmer temperatures in JJA lead to lower GPP simulated by
Farquhar compared to Collatz, and cooler temperatures in MAM result in slightly higher GPP with Farquhar
compared to Collatz (Fig. S5). Using the Medlyn model means simulated $c_i$ is more sensitive to increasing leaf
humidity deficit for the BDT PFT (Fig. S7). Medlyn simulates a lower $c_i$ as humidity deficit increases compared
to Jacobs which leads to lower GPP and LE, the magnitude of which depends on the local site humidity conditions.
In JJA the Medlyn $g_s$ model performs worse at all sites for GPP (Fig. 1b), although improvements in simulated
EF are seen in JJA, where both Medlyn and thermal acclimation improve model performance at three out of four
BDT sites (Fig. 1d, Fig. S6).
At the evergreen needleleaf sites (NET) the most consistent improvements to simulated GPP are seen with the
Farquhar model, where simulated GPP in JJA is substantially improved (GPP reduced) at three out of four sites
(Fig. 1b, Fig. S5), in this season both Medlyn and thermal acclimation generate larger improvements in the
simulated GPP (reducing GPP further), but this is just at two out of the four sites. In our implementation of the
Farquhar model, the NET PFT has a lower $V_{cmax}$ compared to Collatz (Table 2), and a slightly higher $T_{opt}v_{cmax}$
(Table 1, Fig. S1k). The resulting shape of the temperature response curve for photosynthesis (Fig. S1j) means
that at leaf temperatures below ~10°C Farquhar photosynthesis is higher. However above 10 °C Farquhar
photosynthesis is lower compared to Collatz, resulting in simulated GPP in MAM that tends to be higher with
Farquhar than Collatz, and in JJA the opposite occurs (Fig. S5). In MAM and JJA the Medlyn $g_s$ model simulates
some large improvements in EF; $c_i$ simulated by Medlyn is more sensitive to increasing leaf humidity deficit
compared to Jacobs (Fig. S7), which results in lower transpiration and EF, and this compares better to the
observations (Fig. 1, Fig. S6).

**Figure 1.** Relative changes in RMSE for each JULES model configuration compared to Collatz with Jacobs
(Clz.Jac) for hourly daytime a) GPP (March-April-May), b) GPP (June-July-August), c) EF (March-April-May)
and d) EF (June-July-August). Calculated according to equation 11, positive values (reds and yellows) are where
RMSE is lower compared to the Clz.Jac configuration, and therefore indicates an improvement compared to the
Clz.Jac baseline, and the Fluxnet observations. EBF: Broadleaf evergreen tropical tree, GRA: $C_3$ grassland, BDT:
Broadleaf deciduous tree, NET: Needle leaf evergreen tree. The fit of each model configuration to observations
and the RMSE are shown in Fig. S5 (GPP) and Fig. S6 (EF).

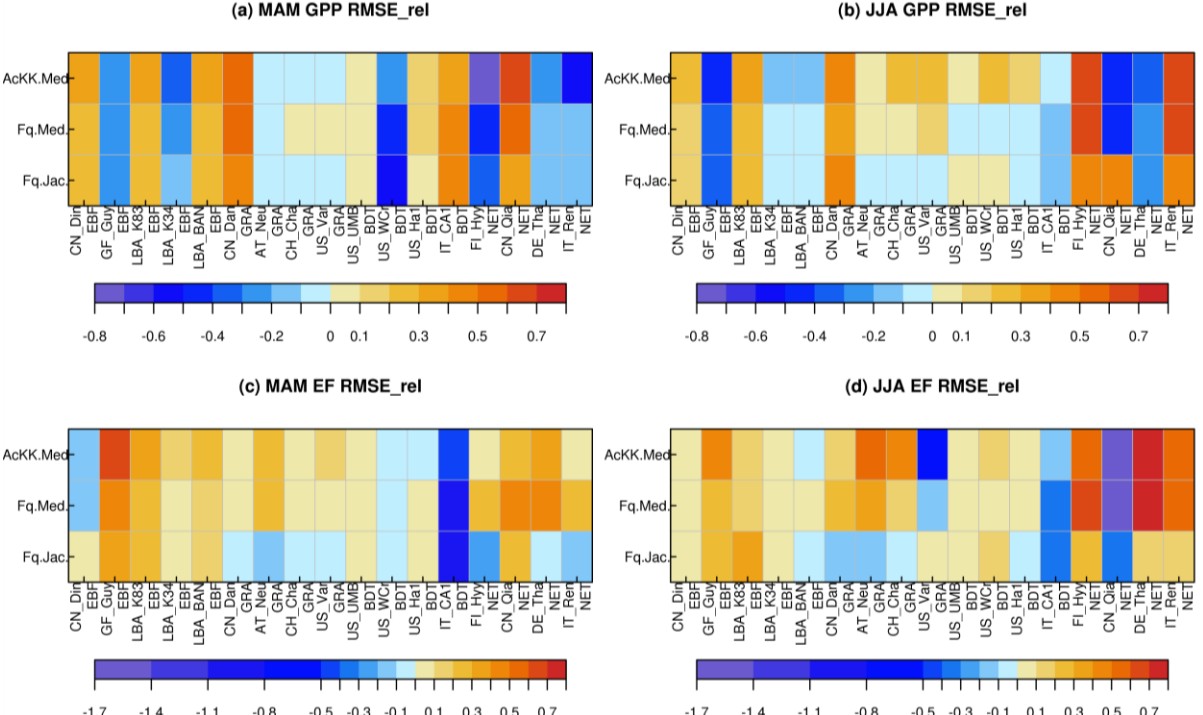





**4.2 Global Evaluation**
**4.2.1 Spatial differences between model configurations**
The impact of changes in the photosynthesis scheme, $g_s$ scheme, adding thermal acclimation of photosynthetic
capacity and the overall change on simulated GPP, LE and H are shown in Figure 2 by comparing each of the new
JULES configurations to the configuration with the alternative process representation. For GPP, the biggest
change is moving from the Collatz photosynthesis scheme to the Farquhar photosynthesis scheme (Fig. 2a). Most
notably, this change results in decreased GPP in the tropical region in JJA of up to 1.5 gC m$^2$ d$^{-1}$ (up to 10%
reduction), whilst in the high northern latitudes, GPP is increased by up to 1.5 gC m$^2$ d$^{-1}$ (up to 20% increase).
This is consistent with results from the site-level simulations where GPP was reduced with implementation of the
Farquhar model at tropical sites, and increased in cooler months (MAM) at the evergreen needleleaf forest sites
(here increased GPP in NET dominated areas are in the forests of the high northern latitudes which is consistent
with cooler temperatures). Impacts on LE and H resulting from the move from Collatz to Farquhar are not as
extensive as those seen with GPP (Fig. 2b & 2c). The change from Jacobs $g_s$ scheme to Medlyn impacts LE and
H most, resulting in a pronounced pattern of decreased LE in northern latitudes (up to 10 W m$^{-2}$, equivalent to a
10% reduction) and corresponding increase in H in JJA (Fig. 2e & 2f). In these JULES simulations, this region is
dominated by NET forest, and the high latitude changes are consistent with results from the site-level simulations,
where using the Medlyn $g_s$ scheme at NET sites resulted in some of the biggest improvements in simulated EF
(lower LE and therefore lower EF). Including thermal acclimation of photosynthesis has the most extensive
impacts on simulated GPP in contrast to LE and H. In the tropical forests GPP is reduced by up to 1 gC m$^2$ d$^{-1}$
(between 2 to 5% reduction) in JJA (Fig. 2g). The impact of acclimation is spatially variable in the temperate
region in JJA, with GPP decreased in Europe (between 2 to 5%), but increased in Eastern USA (up to 20%). Some
areas of the boreal region see increased GPP (between 2 to 5%). This GPP response demonstrates the impact of
thermal acclimation which allows the parameters of the temperature sensitivity functions for photosynthetic
capacity ($V_{cmax}$, $J_{max}$ and $J_{max}$:$V_{cmax}$) to move in response to the temperature of the growth environment, leading to
spatially and temporally different values of the $T_{opt}$ for photosynthesis for each C$_3$ PFT. Thermal acclimation
impacts LE and H to a lesser extent, but where changes are seen, acclimation increases LE with a corresponding
decrease in H (Fig. 2h & 2i). Figs. 2j, 2k & 2l show the overall change that results from moving from the traditional
JULES set-up of Collatz with Jacobs (Clz.Jac) to Farquhar with thermal acclimation and Medlyn $g_s$ (AcKK.Med),
and the impacts on simulated GPP, LE and H can clearly be seen as the trade-off between the dominating effects
from each model configuration. For LE and H the response of the simulated energy fluxes is dominated by the
change in the representation of $g_s$, and for GPP the response of simulated carbon fluxes is dominated by the change
in the representation of photosynthesis and its response to temperature (i.e. thermal acclimation).

**Figure 2.** Absolute difference between JULES modelled GPP, latent (LE) and sensible heat (H) for the different
JULES model configurations in June-July-August (JJA) to show the impact of a, b, c) changing photosynthesis
scheme (Fq.Jac – Clz.Jac); d, e, f) changing $g_s$ scheme (Fq.Med – Fq.Jac); g, h, i) accounting for thermal
acclimation of photosynthesis (AcKK.Med – Fq.Med); and j, k, l) the overall change (AcKK.Med – Clz.Jac),
under present-day meteorological conditions. For each variable the mean over the period 2002 to 2012 is used.
The absolute mean value simulated by each model configuration (JJA) is shown in Fig. S8. DJF is shown in Fig.
S9 (mean absolute values) and Fig. S10 (absolute difference).

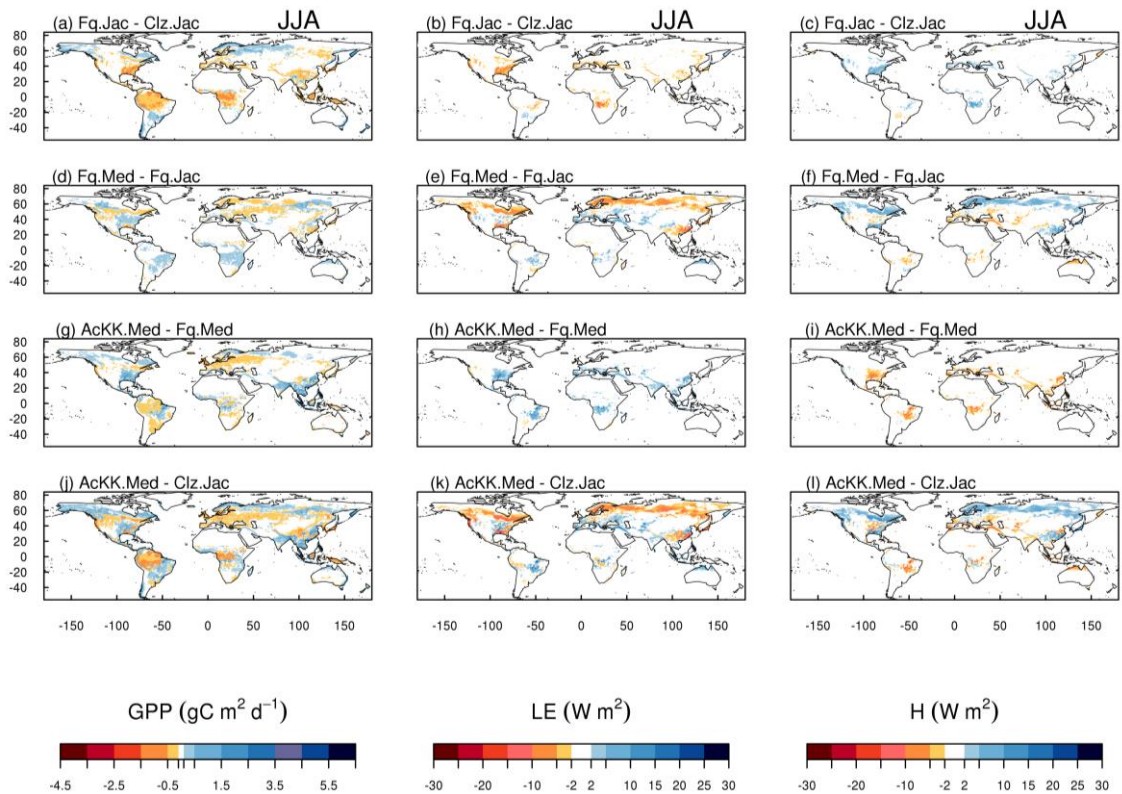

 **4.2.2 Comparison to global estimates: seasonal mean GPP and ET**

Evaluation of simulated global mean GPP by season using FluxCom and MOD17 global GPP products is
presented in Fig. 3a and using global ET from both FluxCom and GLEAM is shown in Fig. 3b. The seasonal
means show thermal acclimation compares best to observations (FluxCom) in JJA (AcKK.Med underestimates
GPP by just 4%, whereas Clz.Jac underestimates GPP by 6%; Fig. 3a & Table S2) and MAM (AcKK.Med
underestimates GPP by just 5%, whereas Clz.Jac underestimates GPP by 11%; Fig. 3a & Table S2), and is in
reasonable agreement with FluxCom in DJF (AcKK.Med overestimates GPP by just 2%, whereas Clz.Jac
underestimates GPP by 4%; Fig. 3a & Table S2). All JULES model configurations have a high GPP bias in SON
compared to FluxCom, and in all seasons GPP is overestimated by all model configurations compared to MOD17,
similarly this is largest in SON. For simulated ET, seasonally the model performance is very similar between the
different JULES configurations, however in both SON and DJF Medlyn (Fq.Med) compares better to both
FluxCom and GLEAM, but the differences are very small (Fig. 3b & Table S3).





**Figure 3.** Seasonal mean global a) GPP and b) ET for each JULES model configuration compared to FluxCom (closed symbols) and MOD17 (GPP) or GLEAM (ET) (open symbols).

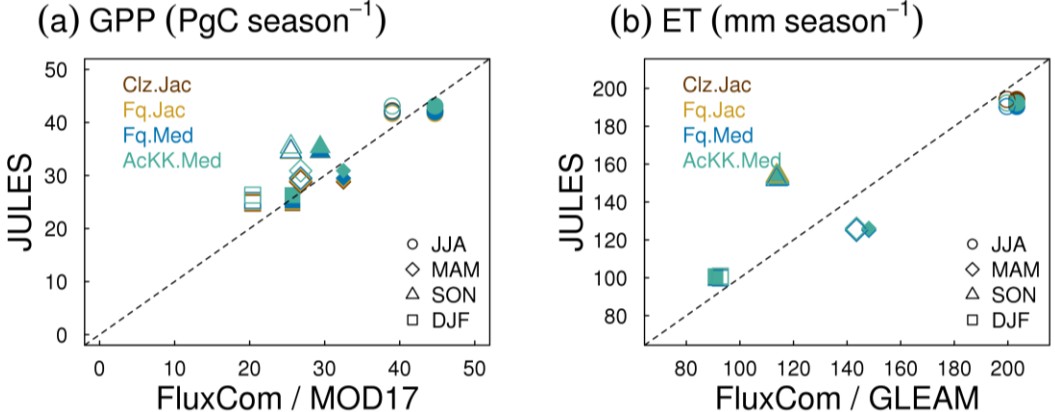

### 4.2.3 Comparison to global estimates: latitudinal mean GPP and ET

Figures 4 and 5 present comparisons of seasonal zonal-mean GPP and ET respectively. Firstly, Fig. 4 and Fig. 5 highlight the differences between global products used to evaluate GPP and ET (see e.g. Spafford & MacDougall 2021). For example, FluxCom generally predicts higher GPP in the tropics compared to MOD17, especially in DJF and MAM, and in JJA the different distribution of GPP by latitude means in the tropics MOD17 GPP is higher than FluxCom in the southern latitudes, and FluxCom GPP is higher in the northern tropics. Comparison of the two ET products shows that GLEAM tends to give higher ET in the tropics, particularly in DJF and MAM. Bearing in mind uncertainties in observation-based estimates of fluxes at this scale we now consider how the different model configurations compare. Notably, all the JULES model configurations in this study simulate comparable global carbon and water fluxes for the recent contemporary period and are in reasonable agreement with the global products used for evaluation. Differences in RMSE between the different model configurations are small for both GPP and ET. Importantly, the most consistent change is the improvement (lowest RMSE) of modelled GPP in the tropics with the Farquhar model (Fq.Jac). This improvement is evident in all seasons and holds when comparing to both FluxCom and MOD17 (Fig. 4). Similarly, estimates of ET are improved in the tropics (lowest RMSE) with the Farquhar model (Fq.Jac) in DJF and JJA, and with the Medlyn model (Fq.Med) in MAM and SON, and again this result is not dependent on the choice of observation-based product (Fig. 5). Another notable change is the improvement of simulated GPP in the temperate north and boreal regions in MAM with thermal acclimation (AcKK.Med). Deficiencies in the model stand out, but these biases are common to all model configurations. For example, all configurations simulate an over-prediction of GPP and ET in SON in the temperate north and boreal regions, overestimated GPP in MAM in tropical southern latitudes (0 to -20ºS), under-predicted GPP and ET in MAM in temperate north and boreal regions, and an over-prediction of ET in MAM in the temperate and tropical South.

**Figure 4.** Mean (2002 to 2012) GPP (g C m² d⁻¹) by latitude band and season for each JULES model configuration
compared to the FluxCom and MOD17 global GPP products. The bars along the side indicate which model
configuration gives the lowest RMSE, and therefore better comparison to FluxCom (righthand bar) and MOD17
(lefthand bar) derived GPP for each region. RMSE values are shown in Tables S4 (FluxCom) and S5 (MOD17).
The grey shaded area shows the uncertainty in the FluxCom GPP product, provided as the median absolute
deviation of ensemble members, this is scaled to a robust estimate of the standard deviation of a normal
distribution by multiplying by 1.4826 according to Jung *et al.*, (2019).

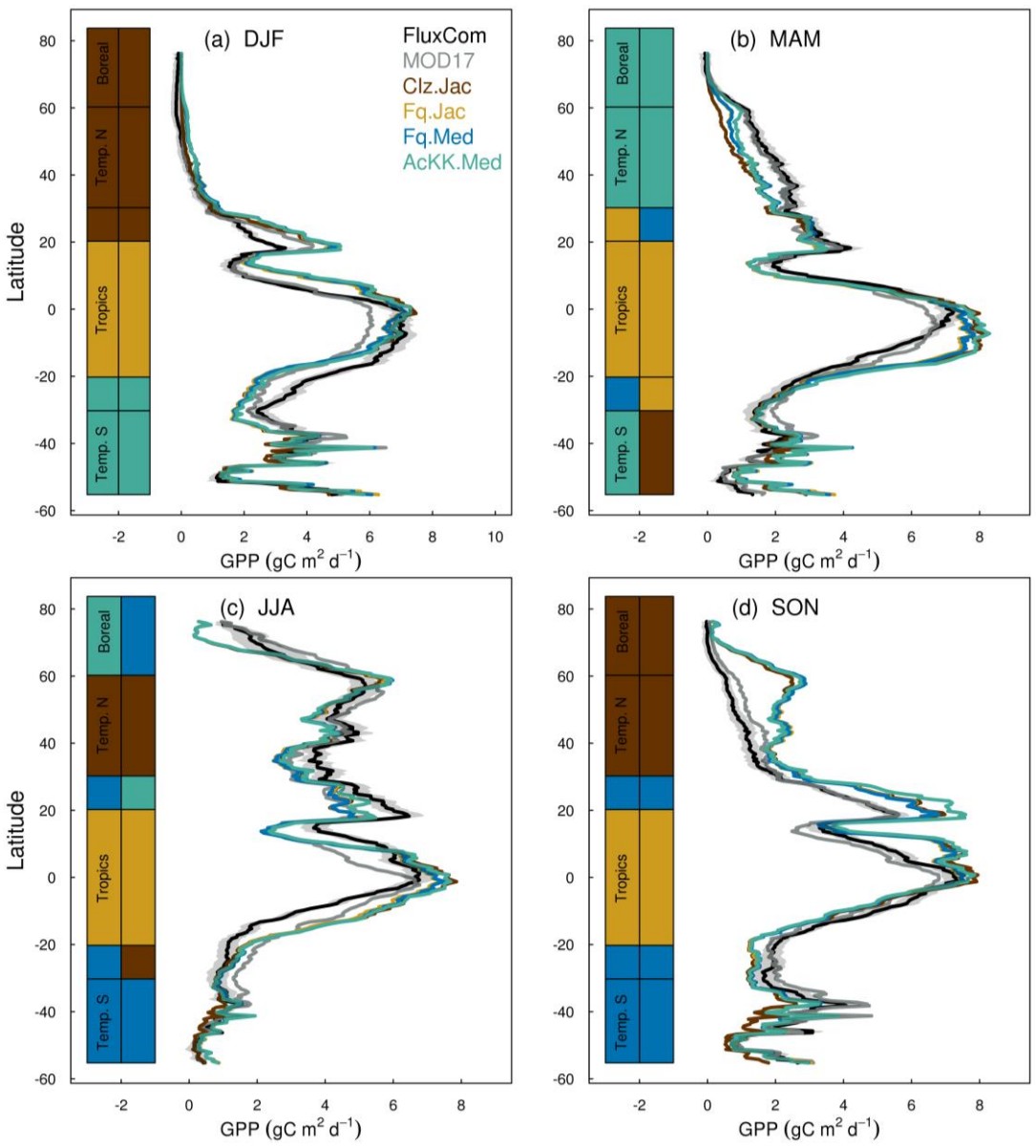






 **Figure 5.** Mean (2002 to 2012) evapotranspiration (ET mm d⁻¹) by latitude band and season for each JULES

model configuration compared to the FluxCom and GLEAM global ET products. The bars along the side indicate

which model configuration gives the lowest RMSE, and therefore better comparison to FluxCom (righthand bar)

and GLEAM (lefthand bar) derived ET for each region. RMSE values are shown in Table S6 (FluxCom) and

Table S7 (GLEAM). The grey shaded area shows the uncertainty in the FluxCom ET product, provided as the

median absolute deviation of ensemble members, this is scaled to a robust estimate of the standard deviation of a

normal distribution by multiplying by 1.4826 according to Jung *et al*., (2019).

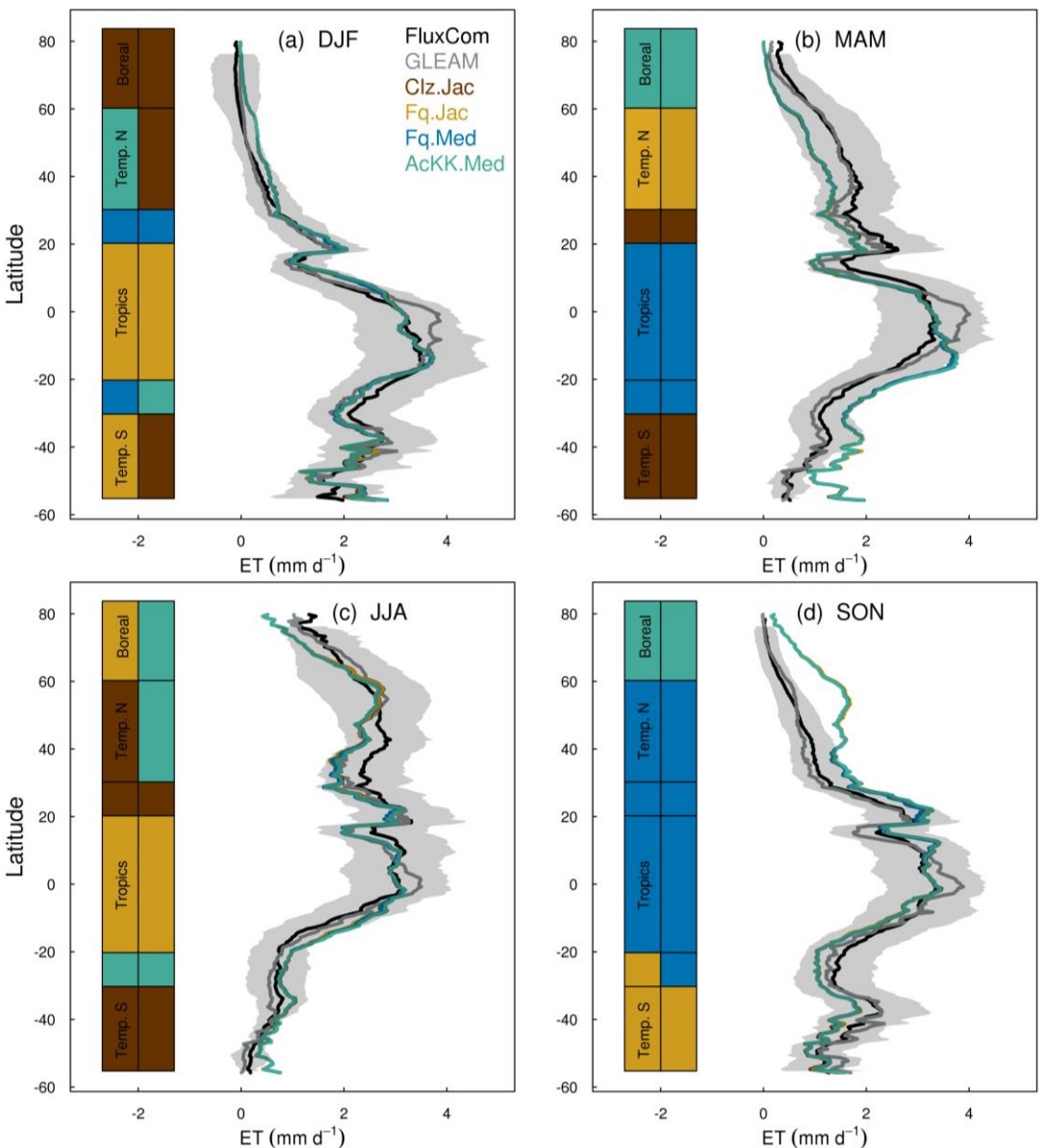

### 4.2.4 Comparison to global estimates: spatial variability of mean GPP and ET

The spatial variability of simulated GPP and ET is shown in Fig. 6 during JJA (Fig. S11 for DJF). We show which

of the JULES model configurations gives the lowest RMSE compared to observation-based estimates of GPP and

ET from FluxCom, MODIS and GLEAM (actual RMSE in Figs. S12 and S13). The differences in RMSE are
typically small between the different JULES model configurations, however some clear patterns emerge. Figure
6a & b show that in the tropical forests of the Amazon basin, central Africa and Southeast Asia (Indonesia, Papua
New Guinea, Malaysia), in both JJA and DJF (Fig. S11a & b for DJF), GPP simulated including thermal
acclimation (AcKK.Med) compares best to both FluxCom and MOD17 across large spatially consistent areas.
Outside of these areas, Fq.Jac also improves the simulation of GPP in the tropics, as does the Medlyn $g_s$ model
(Fq.Med) in JJA in South China and Indo-China. Also, in the high northern latitudes, dominated by evergreen
needleleaf forests, inclusion of thermal acclimation more closely aligns simulated GPP with both FluxCom and
MOD17 (Fig. 6a & b). Compared to FluxCom, ET in JJA is simulated best by thermal acclimation (AcKK.Med)
in the northern temperate and boreal region, although this pattern is not consistent in comparison to GLEAM (Fig.
6c & d). In contrast to GPP, results are more mixed in the tropics for ET. In areas dominated by tropical tree cover,
thermal acclimation (AcKK.Med) and Medlyn (Fq.Med) tend to give the lowest RMSE in JJA and DJF, and in
tropical areas dominated by $C_3$ and $C_4$ grasses Farquhar (Fq.Jac) performs best (Fig. 6c & d), although in DJF the
Medlyn model gives the lowest RMSE in these areas (Fig. S11c & d). In DJF for both GPP and ET, in northern
temperate and boreal regions the Collatz with Jacobs (Clz.Jac) configuration performs the best (Fig. S11).

**Figure 6.** Colours indicate the JULES model configuration that gives the lowest RMSE compared to either the a)
FluxCom and b) MOD17 global GPP (gC m$^2$ day$^{-1}$) products, or c) FluxCom and d) GLEAM global ET (mm day$^{-1}$
$^{1}$) products for JJA over the period 2002 to 2012. Actual RMSE values shown in Fig. S12 and Fig. S13.

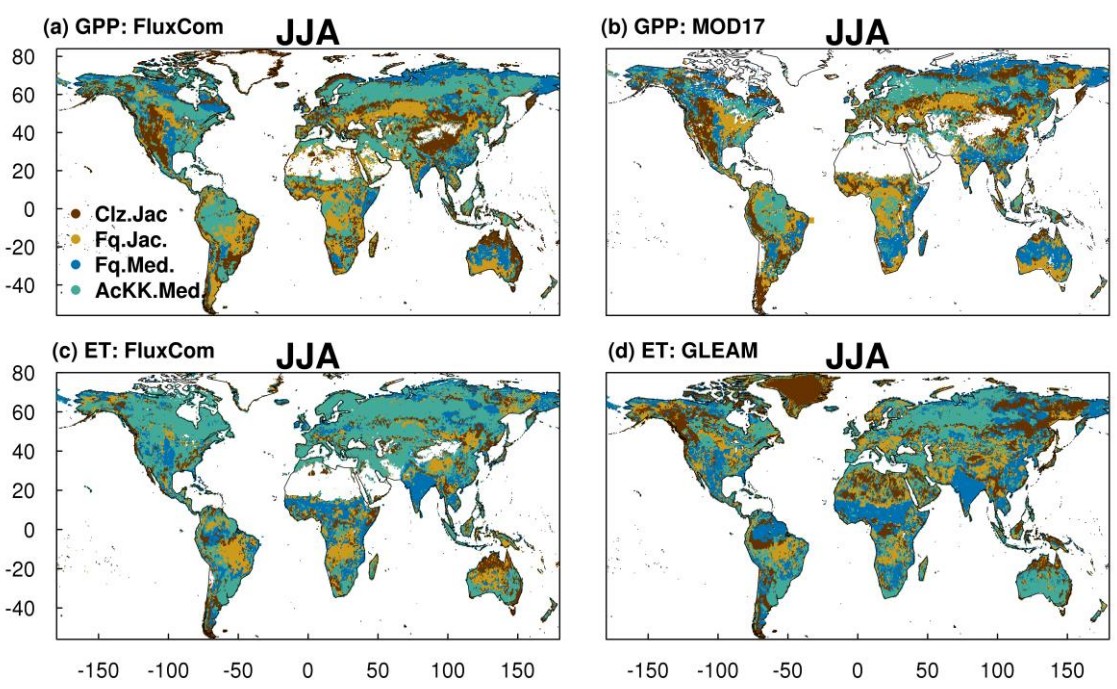



### 4.3 Application under future climate

We run the new configurations forced by variables from a future climate scenario (HadGEM3-GC3.1 forcing
under a high-end emission scenario of the SSPs) to investigate the response of simulated fluxes to long-term
warming. Changing the photosynthesis scheme from Collatz to Farquhar results in lower GPP, (up to 30%
decrease) by 2050 across the high northern latitude forests (Fig. 7a), with the impact on LE (decreased) and H
(increase) less extensive (Fig. 7b & c). This area is dominated by NET, NDT and BDT PFTs in JULES. The
different temperature sensitivity of photosynthesis parameterised with the Farquhar model compared to Collatz
(Fig. S1g, j & m) means at lower leaf temperatures, photosynthesis is higher with Farquhar, however, as leaf
temperature increases, photosynthesis falls in Farquhar relative to Collatz. The crossover point at which this
occurs is relatively low for these PFTs, particularly NET. This impact of the change of temperature sensitivity
was seen in the site-level simulations at FLUXNET NET and BDT sites. There, modelled GPP tended to be higher
with Farquhar than Collatz in MAM, but lower in the warmer conditions of JJA, and in this climate change
scenario the temperate and boreal region both experience large increases in mean annual air temperature ($+5^{\circ}C$
from 1980 to 2060, Fig. S4a & c).
Replacing the Jacobs $g_s$ scheme with Medlyn has the biggest impact on the surface energy fluxes, with increased
LE of up to 30% and a corresponding decrease in H by 2050 across the temperate region (Fig. 7e & f). This area
is dominated by the C3 grass PFT in JULES which has a less conservative water use strategy in the Medlyn
scheme (high $g_1$) compared to Jacobs. This means in the Medlyn scheme, the C3 grass PFT is less sensitive to
increasing humidity deficit at the leaf surface, therefore as humidity deficit increases Medlyn simulates higher $c_i$
leading to higher rate of transpiration and LE compared to Jacobs (Fig. S7).
Thermal acclimation of photosynthesis leads to widespread increases in GPP by 2050 (Fig. 7g). This amounts to
10% in the tropical forests, up to 30% in northern temperate and boreal regions, and up to 40% in south-east Asia.
In this long-term climate change scenario, with large increases in mean annual temperature (Fig. S4), the impact
of thermal acclimation on GPP can clearly be seen. The flexibility in $T_{opt}v$, $T_{opt}j$ and the $J_{max}:V_{cmax}$ ratio of
photosynthesis that thermal acclimation allows through letting these parameters move with the prevailing $T_{growth}$,
allows for higher rates of photosynthesis and therefore GPP as temperatures increase. By contrast, in simulations
where photosynthetic rates are controlled by fixed temperature sensitivies, vegetation may have moved past its
thermal optimum. Time series of the area-weighted mean annual GPP show that in this simulation, across the
tropical region, thermal acclimation enhances GPP by ~7.5 PgC compared to no acclimation (Fig. 8a). In the
temperate region and sub-tropics thermal acclimation increases GPP by ~1 PgC by 2050 (Fig. 8b and d), and in
the boreal region GPP is enhanced by ~0.4 PgC (Fig. 8c). Thermal acclimation of photosynthesis also has a large
impact on simulated energy fluxes, most notably in the northern temperate region, where LE is increased by up
to 50 to 60% (decreased H up to 40 to 50%) (Fig. 7h & i).

**Figure 7.** The difference of difference approach (Equation 12) to determine the impact on GPP (g C $m^2$ $day^{-1}$), LE
and H (both W $m^2$) of the individual changes to each JULES model configuration over the course of the future
(HadGEMGC3.1) simulation (1980 to 2050) in June-July-August (JJA). The AcKK.Med acclimation effect is
calculated from Fig. S16 AcKK.Med – Fq.Med, the effect of the Medlyn $g_s$ scheme is calculated from Fig. S16
Fq.Med – Fq.Jac, and the effect of the photosynthesis scheme is calculated from Fig. S16 Fq.Jac – Clz.Jac.

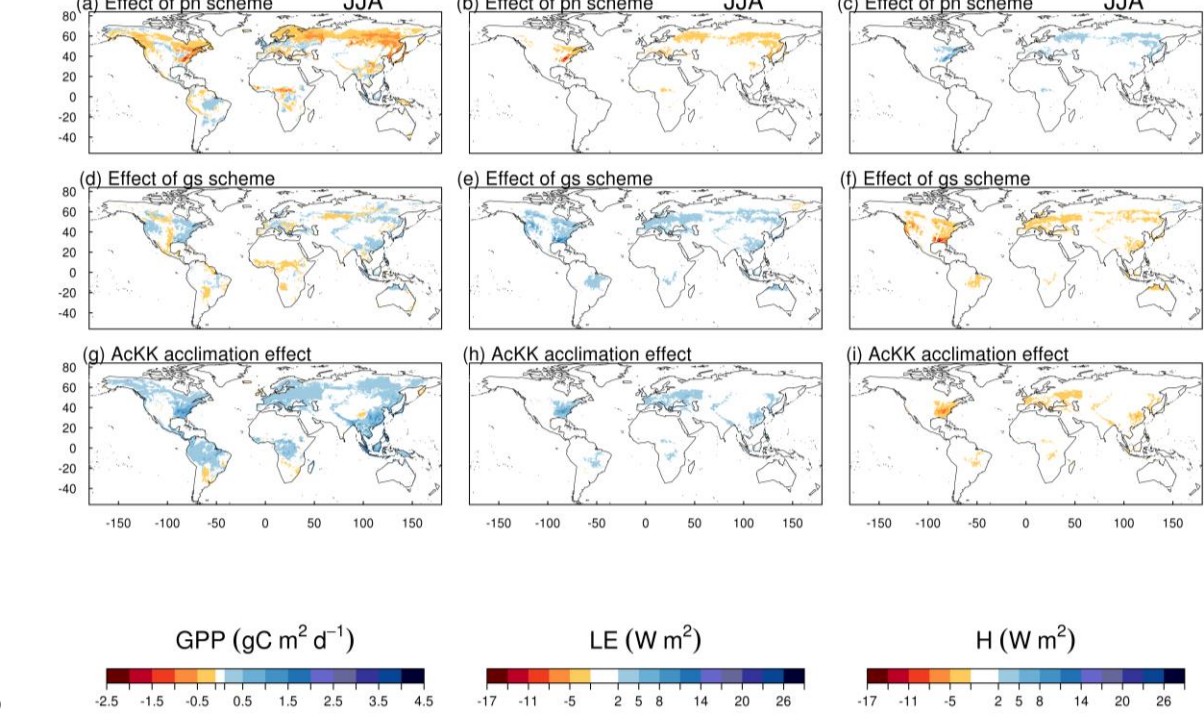



**Figure 8.** Time series of the regional mean acclimation effect i.e AcKK.Med – Fq.Med (black), and the effect of the Medlyn $g_s$ model i.e. Fq.Med – Fq.Jac (grey).

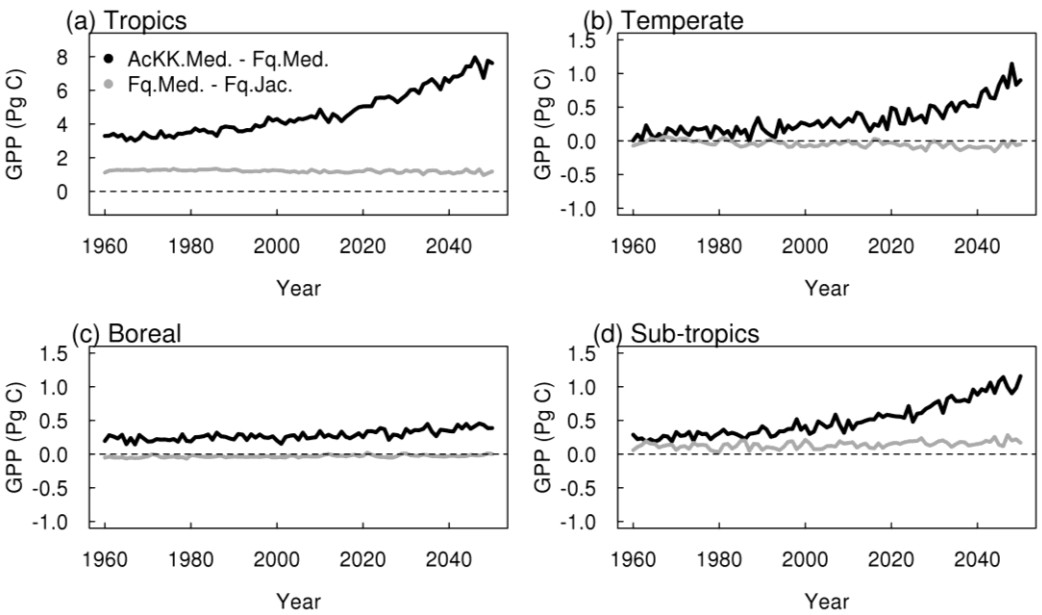


## 5. Discussion

Photosynthesis and $g_s$ are central to the estimate of carbon and water fluxes in LSMs, and when coupled in ESMs these processes feed-back onto the climate system to influence predictions of future climate change. Therefore

improving the representation of these processes in LSMs is important, and previous studies have identified thermal
acclimation of photosynthesis as a key missing process (Booth et al., 2012).

**5.1 Performance of the new JULES plant physiology model configurations: Thermal acclimation**

Our results show that including thermal acclimation of photosynthesis in the JULES model improves simulated
carbon and water fluxes in several key areas for the recent contemporary period. Firstly, the seasonal mean
estimates of global GPP show that in most seasons (JJA, MAM and DJF) thermal acclimation of photosynthesis
with Medlyn $g_s$ (AcKK.Med) predicts GPP in closer agreement with estimates from FluxCom compared to the
traditional 'standard' JULES configuration of Collatz photosynthesis with Jacobs $g_s$ (Clz.Jac). Secondly, thermal
acclimation with Medlyn $g_s$ improves the simulation of GPP (reduces GPP) in the tropical forests in JJA and DJF
(i.e. the Amazon basin and central African rainforest region) and is in closest agreement with estimates of GPP
from both FluxCom and MOD17 for these regions. Thirdly, in the high northern latitude forests dominated by
evergreen needleleaved trees, thermal acclimation increases GPP in JJA and is again in closest agreement with
the observational estimates. Finally, in JJA, AcKK.Med improves the simulation of ET across a large area of the
temperate north and boreal regions.
Our evaluation therefore suggests that fixed, PFT-specific temperature dependencies for $V_{cmax}$ (and $J_{max}$) do not
accurately simulate GPP for the tropical tree and evergreen needleleaf tree PFTs for the present-day in the JULES
model. Thermal acclimation allows the temperature sensitivity of photosynthesis to adjust to the local temperature
environment through flexibility in $T_{opt}v$, $T_{opt}j$ and the $J_{max}$:$V_{cmax}$ ratio. In the tropical forests, for example, GPP is
over-estimated by both Clz.Jac and Fq.Jac. The configuration with thermal acclimation reduces GPP compared to
both these model configurations. From the leaf-level plots in Fig. S1a, the fixed $T_{opt}$ of photosynthesis in the
Collatz scheme is ~33°C and in Farquhar is ~34°C. This is higher than observations from Fig. 1a of Kumarathunge
et al. (2019b), where the $T_{opt}$ for net leaf photosynthesis lies between ~29 to 32°C, and other studies also show a
lower $T_{opt}$ for photosynthesis of around 30°C for mature tropical trees (Hernández et al., 2020; Mau et al., 2018).
This supports our results, and suggests the fixed temperature sensitivity of photosynthesis for tropical trees in the
JULES model results in a $T_{opt}$ of photosynthesis that is too high for current-day. Thermal acclimation results in a
more realistic $T_{opt}$ of photosynthesis for tropical trees because it is influenced by actual growth temperature and
so can adjust to local environmental conditions.
Under the climate change scenario used in this study, thermal acclimation shows a sustained positive acclimation
effect in all regions, increasing GPP in response to long-term warming (although this is less pronounced in the
boreal region). By 2050 GPP was ~10% higher with thermal acclimation in the tropical forests, up to 30 to 40%
higher across a large area of the northern hemisphere. Our findings broadly agree with Mercado et al. (2018), who
implemented the Kattge and Knorr (2007) thermal acclimation scheme into JULES running as part of a coupled
climate-carbon model, and found that thermal acclimation increased land carbon storage in tropical and temperate
regions. This is in contrast to Lombardozzi et al. (2015) and Smith et al. (2016) whose studies both found a
negative impact of photosynthetic thermal acclimation in the tropics, again using the Kattge and Knorr
(2007)thermal acclimation scheme. Mercado et al. (2018) attribute these differences to the method used to
implement acclimation of the $J_{max}$:$V_{cmax}$ ratio at 25°C, that is either reducing $J_{max}$ alone as in the case of the latter

two studies, or by decreasing $J_{max}$ and increasing $V_{cmax}$ simultaneously whilst keeping the total amount of leaf nitrogen the same as used in the present study and in Mercado et al. (2018). The simulated response of thermal acclimation therefore appears to be sensitive to this subtlety in the parameterisation of the acclimation schemes and warrants further investigation. Yet a clear understanding of what drives the change in the $J_{max}$:$V_{cmax}$ ratio in response to $T_{growth}$ is still lacking. More recent results from the analysis by Kumarathunge et al. (2019b) highlight the difficulty in pinning down what drives this process. They found that the $J_{max}$:$V_{cmax}$ ratio responded strongly and consistently to $T_{growth}$, but whether that was achieved by increasing $V_{cmax}$, decreasing $J_{max}$ or both was highly variable.

The behaviour of the thermal acclimation scheme in JULES in response to long term warming implies unlimited thermal resilience of vegetation, but how realistic is this? Observational studies suggest temperate tree species have sufficient capacity to acclimate to rising temperatures e.g. (Drake et al., 2015; Reich et al., 2018; Sendall et al., 2015), although large inter-specific variability in thermal tolerance is identified in co-occurring temperate tree species (Guha et al., 2018). Studies exploring thermal acclimation of photosynthesis for grasslands and C$_3$ herbaceous vegetation are more limited. For boreal tree species, experimental studies suggest high variability between species with respect to photosynthetic acclimation responses to increasing temperatures, for example, there is an increasing body of work suggesting that the evergreen boreal conifer species *Picea* might be particularly vulnerable to warming (Benomar et al., 2017; Dusenge et al., 2020; Kroner and Way, 2016; Kurepin et al., 2018; Way and Sage, 2008; Zhang et al., 2015). The three year open-air warming experiment of Reich et al. (2018) showed that for 11 temperate and boreal tree species studied, warming increased photosynthesis in most species on wet soils, but not in drier conditions. Further, under moist soil conditions, all deciduous species showed an acclimation response to increased temperatures, however, the two boreal evergreen species, *Abies* and *Picea*, showed no thermal acclimation response at any soil moisture concentration. It is generally thought that evergreen species have a reduced capacity to acclimate growth and photosynthesis to warming compared to deciduous tree species (Dusenge et al., 2020; Way and Yamori, 2014). Therefore, the response of boreal forest ecosystems to warming will depend on species composition given the varied acclimation capacities shown and lower diversity of boreal forests, and, as Reich et al. (2018) highlight, also on interaction with other climate changes such as precipitation. In contrast to temperate and boreal forests, tropical forests are thought to be more susceptible to climate change, having evolved under relatively narrow temperature regimes, and experiencing less seasonal and day-to-day variation in temperature changes (Cunningham and Read, 2003). As a consequence, an increasing number of studies show that tropical trees have less capacity to physiologically acclimate photosynthesis to increasing temperatures (Carter et al., 2021; Dusenge et al., 2021; Mau et al., 2018; Miller et al., 2021; Vårhammar et al., 2015). Other studies have determined high temperature threshold responses of photosynthesis, indicating an ability of tropical trees to acclimate to moderate warming, but more severe warming decreases carbon gain (Doughty and Goulden, 2008; Pau et al., 2018; Slot and Winter, 2017; Sullivan et al., 2020). In two tropical understorey species acclimation of the $T_{opt}$ of photosynthesis was observed in the early successional species, whereas no acclimation capacity was shown by the mid-successional species (Carter et al., 2020). Our study demonstrates a large positive impact of thermal acclimation on GPP in tropical forests. However a notable uncertainty in the parameterisation is that the dataset used in the Kattge and Knorr (2007) scheme to construct the empirical relationships is heavily weighted towards temperate species, including only two boreal species and no tropical species (Kattge and Knorr, 2007). There is a significant gap in understanding tropical forest responses to

increasing temperature. Observational studies are starting to address this gap, but this increasing knowledge is yet
to be incorporated into models. Therefore, whilst results from this study demonstrate the importance of thermal
acclimation of photosynthesis on simulation of the future global carbon cycle, they should be interpreted with
some caution. The varied results from experimental studies highlights the research needed to further understand
thermal acclimation responses in a variety of ecosystems, over different timescales, and from leaf-level through
to canopy, and finally to translate that understanding so it is amenable to incorporation into ESMs.
**5.2 Performance of the new JULES plant physiology model configurations: Medlyn $g_s$**
In this study, the Medlyn $g_s$ model had the biggest impact on surface energy fluxes simulated by the $C_3$ grass PFT
and needleleaf evergreen tree PFT in JULES. This reflects a change to the water-use strategy of these PFTs as
reported by Lin et al. (2015) that is not currently captured by parameterisations in the JULES Jacobs model. Global
simulations with the Medlyn scheme for the recent contemporary period simulated a ~10% decrease in LE
(increased H) across the high northern latitudes dominated by the NET PFT compared to the standard JULES
Jacobs $g_s$ scheme. The future climate change experiment showed a large response across the temperate region
dominated by the C3 PFT, where LE increased by ~30% (H decreased) with Medlyn. Our study for current-day
is in agreement with De Kauwe et al. (2015) who found a large impact of the Medlyn model on transpiration
fluxes in needle leaved evergreen trees (~30% reduction) in the CABLE LSM . Coupled simulations using CABLE
within the Australian Community Climate and Earth Systems Simulator (ACCESSv1.3b) showed that the Medlyn
$g_s$ scheme reduced the LE flux from the land surface over the boreal forests during JJA by 0.5–1.0 mm day$^{-1}$,
leading to warmer daily maximum and minimum temperatures by up to 1.0°C and warmer extreme maximum
temperatures by up to 1.5°C (Kala et al., 2015). In future simulations, this new parameterisation of the stomatal
scheme in ACCESS1.3 substantially increased the intensity of future heatwaves across Northern Eurasia (Kala et
al., 2016).
**5.3 Implications for land-atmosphere feedbacks**
Modifying the leaf-level stomatal behaviour in JULES impacts the simulated surface energy fluxes. In our study,
a change of stomatal opening results from either a direct change in the parameterisation of $g_s$ or through altered
stomatal behaviour in response to temperature. In our offline climate change simulation, thermal acclimation
increased stomatal opening in response to long term warming, and in some regions this increased the rate of
transpiration and evaporative cooling, and decreased the sensible heat flux. When coupled to an atmospheric
model, such behaviours have potential to feed-back on the land surface via changes in temperature, cloud cover
and precipitation, as for example modelled by De Arellano et al. (2012); Kala et al. (2015); Kala et al. (2016);
Kooperman et al. (2018); Zeng et al. (2017). The extent and amplitude of acclimation-induced perturbations to
surface energy fluxes in our offline simulation suggests a potential impact on regional scale circulations, for
example across the East Asian monsoon region. The impact of these changes to the plant physiology routines in
JULES on land-atmosphere feedbacks will be investigated in future work through coupled simulations in the
HadGEM global climate model.
**5.4 Limitations of this study**
Across all latitudes, the changes introduced to JULES by the new plant physiology routines did not degrade the
performance of JULES. All model configurations compared reasonably well to the FluxCom and MOD17 GPP

products, and FluxCom and GLEAM ET products, given that there are also uncertainties inherent in estimates from these products. For example, the satellite-based products of GPP have recently been shown to incorrectly capture the response of photosynthesis to $CO_2$, which means they potentially underestimate the response of GPP to rising atmospheric $CO_2$ (Keenan et al., 2021). Nevertheless, some notable biases in the model were identified that were common to all JULES model configurations, for example the over-prediction of GPP and ET in the temperate and boreal region in SON, and the over-prediction of both fluxes in MAM in the southern tropics (0 to -20°S). Potential sources of error to consider may be the use of a prescribed climatology of MODIS based LAI, which some studies have reported to be inaccurate over forested areas (Shabanov et al., 2005). Other processes currently missing in the model may also contribute to these large biases, such as a lack of seasonality in photosynthetic capacity (i.e. $V_{cmax}$ and $J_{max}$) which has been demonstrated for many different forest species (Croft et al., 2017; Wilson et al., 2001), and without which likely causes over-estimation of forest carbon exchange. For example, in SON the high GPP and ET bias occurs in the northern temperate and boreal region which could be linked to a lack of photosynthetic phenology in the model. Towards the end of the growing season leaves in this region have reduced nitrogen content and therefore lower photosynthetic capacity, but because JULES uses a fixed value for photosynthetic capacity JULES maintains a high rate of carbon assimilation despite having seasonal LAI.

More generally, this study revealed limited data to inform the temperature sensitivity response functions of different PFTs for implementation into LSMs. We found only a few datasets for $C_3$ grass/herbaceous vegetation (e.g. Wohlfahrt *et al*., (1999) and Joseph *et al*., (2014)) which represents only limited geographical coverage. Consequently, we fitted the temperature response function for this PFT in the Farquhar scheme to that of the existing function in the JULES Collatz photosynthesis scheme. We also encountered an issue regarding uncertainty about the temperature response functions at low temperatures. The data-led functions we implemented for all PFTs (with the exception of the C3 PFT) from Kumarathunge et al. (2019b) showed higher rates of leaf-level photosynthesis at low leaf temperatures compared to the existing functions in the JULES Collatz scheme, where photosynthesis was much lower and goes to zero at 0 °C for most PFTs (see PFT leaf-level temperature sensitivity curves for gross photosynthesis in Fig. S1). In our simulations this led to higher GPP in DJF when using the Farquhar scheme, which increased biases with respect to FluxCom and MOD17 global estimates of GPP. It is desirable to use the temperature response functions from Kumarathunge et al. (2019b) as these are entirely data-led. However for some PFTs the resulting behaviour of photosynthesis at very low temperatures looks potentially unrealistic, and the question here is how well constrained by observations are the temperature sensitivity curves at low temperatures? For global modelling applications, understanding the response of photosynthesis to temperature over a wide temperature range is essential, including at low temperatures as well as around the $T_{opt}$ of photosynthesis for different species and PFTs. Additionally, increasing the understanding and data availability of the temperature sensitivity of different species from different biomes will allow greater representation within LSMs of the variation that exists across the globe.

The simulations presented in this work use a prescribed map of vegetation cover which means the extent and location of each PFT does not change over time. The model can alternatively be run with dynamic vegetation enabled, which means the model predicts the extent of each PFT, and therefore vegetation cover can change in space and time as PFTs compete with each other in response to changing climatic conditions. Yet to be explored

as part of this work, is how changes to the plant physiology routines, as implemented here, might affect the extent
of different PFTs over time when vegetation dynamics is enabled. For example, changes to the temperature
response of photosynthesis may lead to a competitive advantage of one PFT over another, and therefore the
vegetation distribution may be very different as temperatures rise compared to simulations that either use the
original Collatz temperature sensitivities or do not include thermal acclimation of photosynthesis. We hypothesise,
for example, that allowing thermal acclimation of the temperature sensitivity of photosynthesis would make the
vegetation distribution more stable in a warmer climate as vegetation can adjust its photosynthetic capacity to
function more efficiently as temperatures rise. Applied in a coupled ESM, a change in vegetation distribution
would impact projections of future climate change.
The treatment of soil moisture stress in JULES is through a linear response function (the $\beta$ function, Eq. 12 in
Best et al., 2011), the use of which in JULES and other LSMs has been identified as a key source of uncertainty
(Blyth et al., 2011; Verhoef and Egea, 2014; Vidale et al., 2021). Incorrect representation of soil moisture stress
has large impacts for modelled carbon and water fluxes, and is of particular importance as droughts are predicted
to increase in frequency or intensity in the future. Work is ongoing to improve the representation of soil moisture
stress in JULES. Harper et al. (2021) investigated alternative parameterisations for $\beta$ and found that increasing
modelled soil depth and therefore plant access to deep soil moisture improved the simulation of soil moisture
stress at eddy covariance flux tower sites. In addition, using soil matric potential instead of volumetric water
content in the $\beta$ function allowed for PFT specific parameterisation of soil moisture stress responses to further
improve modelled fluxes. Vidale et al. (2021) explored combinations of non-linear $\beta$ function responses applied
at different points in the photosynthesis – $g_s$ pathway (i.e. carbon assimilation, $g_s$, or mesophyll conductance).
They found that treatments allowing $\beta$ to act on vegetation fluxes via stomatal and mesophyll routes were able to
better capture the spatiotemporal variability in water use efficiency during the growing season. However, in
addition to these alternative parameterisations of $\beta$, further developments to how the soil-plant hydraulic system
is represented in JULES are being made, including an optimality based plant hydraulic transport model recently
implemented in JULES (Eller et al., 2020).
Whilst the development of multi-layer canopy radiation models in LSMs has improved the simulation of radiation
and energy within vegetation canopies, the interception of light by plants in JULES, like most LSMs, is not well
represented despite being critical to predicting the uptake of carbon by plants (Loew et al., 2014). LSMs generally
make the simplifying assumption that leaves are randomly arranged in space, instead of being clustered into tree
crowns or around branches, leaving gaps in and around the canopy. Shortwave radiation is used by plants to
photosynthesise, and canopy structure has a direct impact on the fraction of this radiation absorbed. Therefore
canopy architecture plays an important role in the partitioning of incident solar radiation,
photosynthesis, transpiration and momentum fluxes (Braghiere et al., 2019). More recently, alternative
approaches are being considered to represent the forest light environment in LSMs to account for the structural
effects of vegetation on radiation partitioning, ranging from canopy clumping parameterisations (Braghiere et al.,
2019; Braghiere et al., 2020; Braghiere et al., 2021) to 3-dimensional models of the canopy light environment
(Hogan et al., 2018; Kobayashi et al., 2012), embedded in radiative transfer schemes, although the latter tend to
be computationally expensive (Yang et al., 2001). Braghiere et al. (2019) incorporated canopy clumping from
satellite data into JULES which resulted in an increase in carbon uptake by photosynthesis. The greatest effect
were in the tropics, where the canopy clumping parameterisation allowed more light to reach the lower layers of
the canopy where photosynthesis tends to be limited by light availability.

**5.5 Conclusions**
Here we introduce new representations of plant physiological processes into the JULES model, building enhanced
capability, and allowing stronger links between model and field studies. This work a) introduces updated
understanding of plant physiological processes into JULES, b) increases the flexibility of the modelling capacity
within JULES by allowing use of two alternative photosynthesis and $g_s$ schemes, in addition to thermal
acclimation of photosynthesis, and c) provides new parameters that are entirely based on large observational
datasets. Testing and evaluation at site-level and globally show some key improvements are made to the JULES
model. Thermal acclimation of photosynthesis coupled with the optimality-based $g_s$ scheme led to improved
simulated carbon fluxes across much of the tropics for the present-day. With about 40% of the world's vegetation
carbon residing in tropical forests, they play a crucial role in regulating both regional and global climate through
water and carbon cycle dynamics (Erb et al., 2018; Pan et al., 2011). Therefore, accurate representation of tropical
carbon fluxes within LSMs is important. Thermal acclimation and the optimality-based $g_s$ scheme also improved
simulated carbon fluxes in the high northern latitude forests in the northern hemisphere summer, and the same
model configuration also improved simulated water fluxes across much of this region in the same season. The
optimality-based Medlyn $g_s$ scheme reduced the LE flux substantially across the northern boreal forests in JJA.
This change reflects a more conservative water-use strategy for the needleleaf evergreen tree PFT that dominates
in this region as suggested by the global synthesis of experimental data from Lin et al. (2015). The current JULES
Jacobs scheme parameterisation does not accurately capture the water-use strategy of this PFT. Our future climate
experiment highlights the impact of thermal acclimation on simulating carbon cycle dynamics and energy fluxes
in response to long-term warming. The potential impact of this altered stomatal behaviour on land-atmosphere
feedbacks via changes in surface energy fluxes will be examined in future coupled simulations.

**Code/Data availability**
JULES-vn5.6 was used for all simulations. The JULES model code and suites used to run the model are available
from the Met Office Science Repository Service (MOSRS). Registration is required and code is freely available
to anyone for non-commercial use (see here for details of licensing https://jules.jchmr.org/content/code). Visit the
JULES website (https://jules.jchmr.org/content/getting-started) to register for a MOSRS account. The results
presented in this paper were obtained by running JULES from the following branch:
https://code.metoffice.gov.uk/trac/jules/browser/main/branches/dev/douglasclark/vn5.6_acclimation@16578.
This is a development branch of JULES-vn5.6 to include thermal acclimation of photosynthesis as described in
this paper. This branch can be accessed and downloaded from the Met Office Science Repository Service once
the user has registered for an account, as outlined above. Documentation for the JULES model is located here:
https://jules-lsm.github.io/vn5.6/. Output data from the model simulations, and R scripts to produce the plots in
the paper are provided at (https://doi.org/10.5281/zenodo.5825540). Site-level simulations used the rose suite u-
br064 (https://code.metoffice.gov.uk/trac/roses-u/browser/b/r/0/6/4/ at revision 146216) which is a copy of the u-
al752 JULES suite for FLUXNET 2015 and LBA sites described here
https://code.metoffice.gov.uk/trac/jules/wiki/FluxnetandLbaSites, and downloaded from here
https://code.metoffice.gov.uk/trac/roses-u/browser/a/l/7/5/2/ at revision 145397). The global simulations used
JULES rose suite u-bq898 (https://code.metoffice.gov.uk/trac/roses-u/browser/b/q/8/9/8/ at revision 181188)
which uses the Global Land configuration 7.1 (Wiltshire et al., 2020). Suites can be downloaded from MOSRS
once the user has registered for an account.
**Competing Interests**
The authors declare no competing interests.
**Author Contributions**
RJO performed simulations and analysis and wrote the first version of the manuscript. DBC, LMM and RJO
developed the model. PLV, PCM and MT provided data for the future climate runs, help with developing the
JULES suites, and general expertise. CH assisted with analysis. SF and SS provided ancillary data for forcing the
model. LMM, CMT, CH, PLV, BEM, PCM and MT contributed to editing the manuscript. All authors contributed
to discussions throughout to develop the work.
**Acknowledgements**
This work and its contributors (RJO, LMM, CH, CMT, PLV, PCM, MT and SF) were supported by the UK-China
Research & Innovation Partnership Fund through the Met Office Climate Science for Service Partnership (CSSP)
China as part of the Newton Fund. R.J.O. acknowledges support from the Natural Environment Research Council,
grant NEC05816 LTS-M-UKESM. This work used eddy covariance data acquired and shared by the FLUXNET
community, including these networks: AmeriFlux, AfriFlux, AsiaFlux, CarboAfrica, CarboEuropeIP, CarboItaly,
CarboMont, ChinaFlux, Fluxnet-Canada, GreenGrass, ICOS, KoFlux, LBA, NECC, OzFlux-TERN, TCOS-
Siberia and USCCC. The ERA-Interim reanalysis data are provided by ECMWF and processed by LSCE. The
FLUXNET eddy covariance data processing and harmonization was carried out by the European Fluxes Database
Cluster, AmeriFlux Management Project, and Fluxdata project of FLUXNET, with the support of CDIAC and
ICOS Ecosystem Thematic Center, and the OzFlux, ChinaFlux and AsiaFlux offices.

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

Contributions of carbon cycle uncertainty to future climate projection spread, Tellus B: Chemical and Physical
Meteorology, 61, 355-360, 10.1111/j.1600-0889.2008.00414.x, 2009.

Jacobs, C. M. J.: Direct impact of atmospheric $CO_2$ enrichment on regional transpiration, Jacobs, S.l., 1994.

Jarvis, P. G., Monteith, J. L., and Weatherley, P. E.: The interpretation of the variations in leaf water potential
and stomatal conductance found in canopies in the field, Philosophical Transactions of the Royal Society of
London. B, Biological Sciences, 273, 593-610, doi:10.1098/rstb.1976.0035, 1976.

Jasechko, S., Sharp, Z. D., Gibson, J. J., Birks, S. J., Yi, Y., and Fawcett, P. J.: Terrestrial water fluxes
dominated by transpiration, Nature, 496, 347-350, 10.1038/nature11983, 2013.

Jogireddy, V. R., Cox, P. M., Huntingford, C., Harding, R. J., and Mercado, L. M.: An improved description of
canopy light interception for use in a GCM land-surface scheme: calibration and testing against carbon fluxes at
a coniferous forest, Hadley Centre Technical Note 63, Hadley Centre, Met Office, Exeter, UK, 2006.

Joseph, T., Whitehead, D., and Turnbull, M. H.: Soil water availability influences the temperature response of
photosynthesis and respiration in a grass and a woody shrub, Functional Plant Biology, 41, 468-481,
10.1071/FP13237, 2014.
Jung, M., Koirala, S., Weber, U., Ichii, K., Gans, F., Camps-Valls, G., Papale, D., Schwalm, C., Tramontana,
G., and Reichstein, M.: The FLUXCOM ensemble of global land-atmosphere energy fluxes, Scientific Data, 6,
74, 10.1038/s41597-019-0076-8, 2019.
Jung, M., Schwalm, C., Migliavacca, M., Walther, S., Camps-Valls, G., Koirala, S., Anthoni, P., Besnard, S.,
Bodesheim, P., Carvalhais, N., Chevallier, F., Gans, F., Goll, D. S., Haverd, V., Köhler, P., Ichii, K., Jain, A. K.,
Liu, J., Lombardozzi, D., Nabel, J. E. M. S., Nelson, J. A., O'Sullivan, M., Pallandt, M., Papale, D., Peters, W.,
Pongratz, J., Rödenbeck, C., Sitch, S., Tramontana, G., Walker, A., Weber, U., and Reichstein, M.: Scaling
carbon fluxes from eddy covariance sites to globe: synthesis and evaluation of the FLUXCOM approach,
Biogeosciences, 17, 1343-1365, 10.5194/bg-17-1343-2020, 2020.
Kala, J., De Kauwe, M. G., Pitman, A. J., Medlyn, B. E., Wang, Y.-P., Lorenz, R., and Perkins-Kirkpatrick, S.
E.: Impact of the representation of stomatal conductance on model projections of heatwave intensity, Scientific
Reports, 6, 23418, 10.1038/srep23418, 2016.
Kala, J., De Kauwe, M. G., Pitman, A. J., Lorenz, R., Medlyn, B. E., Wang, Y. P., Lin, Y. S., and Abramowitz,
G.: Implementation of an optimal stomatal conductance scheme in the Australian Community Climate Earth
Systems Simulator (ACCESS1.3b), Geosci. Model Dev., 8, 3877-3889, 10.5194/gmd-8-3877-2015, 2015.
Kattge, J. and Knorr, W.: Temperature acclimation in a biochemical model of photosynthesis: a reanalysis of
data from 36 species, Plant, Cell & Environment, 30, 1176-1190, 10.1111/j.1365-3040.2007.01690.x, 2007.
Keenan, T. F., Luo, X., De Kauwe, M. G., Medlyn, B. E., Prentice, I. C., Stocker, B. D., Smith, N. G., Terrer,
C., Wang, H., Zhang, Y., and Zhou, S.: A constraint on historic growth in global photosynthesis due to
increasing $CO_2$, Nature, 600, 253-258, 10.1038/s41586-021-04096-9, 2021.
Kobayashi, H., Baldocchi, D. D., Ryu, Y., Chen, Q., Ma, S., Osuna, J. L., and Ustin, S. L.: Modeling energy and
carbon fluxes in a heterogeneous oak woodland: A three-dimensional approach, Agricultural and Forest
Meteorology, 152, 83-100, 2012.
Kooperman, G. J., Chen, Y., Hoffman, F. M., Koven, C. D., Lindsay, K., Pritchard, M. S., Swann, A. L. S., and
Randerson, J. T.: Forest response to rising $CO_2$ drives zonally asymmetric rainfall change over tropical land,
Nature Climate Change, 8, 434-440, 10.1038/s41558-018-0144-7, 2018.
Krinner, G., Viovy, N., de Noblet-Ducoudré, N., Ogée, J., Polcher, J., Friedlingstein, P., Ciais, P., Sitch, S., and
Prentice, I. C.: A dynamic global vegetation model for studies of the coupled atmosphere-biosphere system,
Global Biogeochemical Cycles, 19, 10.1029/2003GB002199, 2005.
Kroner, Y. and Way, D. A.: Carbon fluxes acclimate more strongly to elevated growth temperatures than to
elevated $CO_2$ concentrations in a northern conifer, Global Change Biology, 22, 2913-2928, 10.1111/gcb.13215,
1244 2016.
Kumarathunge, D. P., Medlyn, B. E., Drake, J. E., Rogers, A., and Tjoelker, M. G.: No evidence for triose
phosphate limitation of light-saturated leaf photosynthesis under current atmospheric $CO_2$ concentration, Plant,
Cell & Environment, 42, 3241-3252, 10.1111/pce.13639, 2019a.
Kumarathunge, D. P., Medlyn, B. E., Drake, J. E., Tjoelker, M. G., Aspinwall, M. J., Battaglia, M., Cano, F. J.,
Carter, K. R., Cavaleri, M. A., Cernusak, L. A., Chambers, J. Q., Crous, K. Y., De Kauwe, M. G., Dillaway, D.
N., Dreyer, E., Ellsworth, D. S., Ghannoum, O., Han, Q., Hikosaka, K., Jensen, A. M., Kelly, J. W. G., Kruger,
E. L., Mercado, L. M., Onoda, Y., Reich, P. B., Rogers, A., Slot, M., Smith, N. G., Tarvainen, L., Tissue, D. T.,
Togashi, H. F., Tribuzy, E. S., Uddling, J., Vårhammar, A., Wallin, G., Warren, J. M., and Way, D. A.:
Acclimation and adaptation components of the temperature dependence of plant photosynthesis at the global
scale, New Phytologist, 222, 768-784, 10.1111/nph.15668, 2019b.

Kurepin, L. V., Stangl, Z. R., Ivanov, A. G., Bui, V., Mema, M., Hüner, N. P. A., Öquist, G., Way, D., and
Hurry, V.: Contrasting acclimation abilities of two dominant boreal conifers to elevated $CO_2$ and temperature,
Plant, Cell & Environment, 41, 1331-1345, 10.1111/pce.13158, 2018.
Leuning, R.: A critical appraisal of a combined stomatal-photosynthesis model for $C_3$ plants, Plant, Cell &
Environment, 18, 339-355, 10.1111/j.1365-3040.1995.tb00370.x, 1995.
Lin, Y.-S., Medlyn, B. E., Duursma, R. A., Prentice, I. C., Wang, H., Baig, S., Eamus, D., de Dios, Victor R.,
Mitchell, P., Ellsworth, D. S., de Beeck, M. O., Wallin, G., Uddling, J., Tarvainen, L., Linderson, M.-L.,
Cernusak, L. A., Nippert, J. B., Ocheltree, T. W., Tissue, D. T., Martin-StPaul, N. K., Rogers, A., Warren, J. M.,
De Angelis, P., Hikosaka, K., Han, Q., Onoda, Y., Gimeno, T. E., Barton, C. V. M., Bennie, J., Bonal, D., Bosc,
A., Löw, M., Macinins-Ng, C., Rey, A., Rowland, L., Setterfield, S. A., Tausz-Posch, S., Zaragoza-Castells, J.,
Broadmeadow, M. S. J., Drake, J. E., Freeman, M., Ghannoum, O., Hutley, Lindsay B., Kelly, J. W., Kikuzawa,
K., Kolari, P., Koyama, K., Limousin, J.-M., Meir, P., Lola da Costa, A. C., Mikkelsen, T. N., Salinas, N., Sun,
W., and Wingate, L.: Optimal stomatal behaviour around the world, Nature Climate Change, 5, 459-464,
10.1038/nclimate2550, 2015.
Liu, H., Randerson, J. T., Lindfors, J., Massman, W. J., and Foken, T.: Consequences of Incomplete Surface
Energy Balance Closure for CO2 Fluxes from Open-Path CO2/H2O Infrared Gas Analysers, Boundary-Layer
Meteorology, 120, 65-85, 10.1007/s10546-005-9047-z, 2006.
Loew, A., Van Bodegom, P., Widlowski, J.-L., Otto, J., Quaife, T., Pinty, B., and Raddatz, T.: Do we (need to)
care about canopy radiation schemes in DGVMs? Caveats and potential impacts, Biogeosciences, 11, 1873-
1281    1897, 2014.

Lombardozzi, D. L., Bonan, G. B., Smith, N. G., Dukes, J. S., and Fisher, R. A.: Temperature acclimation of
photosynthesis and respiration: A key uncertainty in the carbon cycle-climate feedback, Geophysical Research
Letters, 42, 8624-8631, 10.1002/2015GL065934, 2015.
Mau, A. C., Reed, S. C., Wood, T. E., and Cavaleri, M. A.: Temperate and Tropical Forest Canopies are
Already Functioning beyond Their Thermal Thresholds for Photosynthesis, Forests, 9, 47, 2018.
Medlyn, B. E., Loustau, D., and Delzon, S.: Temperature response of parameters of a biochemically based
model of photosynthesis. I. Seasonal changes in mature maritime pine (Pinus pinaster Ait.), Plant, Cell &
Environment, 25, 1155-1165, 10.1046/j.1365-3040.2002.00890.x, 2002.
Medlyn, B. E., Duursma, R. A., Eamus, D., Ellsworth, D. S., Prentice, I. C., Barton, C. V. M., Crous, K. Y., de
Angelis, P., Freeman, M., and Wingate, L.: Reconciling the optimal and empirical approaches to modelling
stomatal conductance, Global Change Biology, 17, 2134-2144, 10.1111/j.1365-2486.2010.02375.x, 2011.
Meir, P., Kruijt, B., Broadmeadow, M., Barbosa, E., Kull, O., Carswell, F., Nobre, A., and Jarvis, P. G.:
Acclimation of photosynthetic capacity to irradiance in tree canopies in relation to leaf nitrogen concentration
and leaf mass per unit area, Plant, Cell & Environment, 25, 343-357, 10.1046/j.0016-8025.2001.00811.x, 2002.
Mercado, L. M., Huntingford, C., Gash, J. H. C., Cox, P. M., and Jogireddy, V. R.: Improving the representation
of radiation interception and photosynthesis for climate model applications, Tellus B, 59, 553-565,
10.1111/j.1600-0889.2007.00256.x, 2007.
Mercado, L. M., Bellouin, N., Sitch, S., Boucher, O., Huntingford, C., Wild, M., and Cox, P. M.: Impact of
changes in diffuse radiation on the global land carbon sink, Nature, 458, 1014-1017, 10.1038/nature07949,
1308    2009.

Mercado, L. M., Medlyn, B. E., Huntingford, C., Oliver, R. J., Clark, D. B., Sitch, S., Zelazowski, P., Kattge, J.,
Harper, A. B., and Cox, P. M.: Large sensitivity in land carbon storage due to geographical and temporal
variation in the thermal response of photosynthetic capacity, New Phytologist, 218, 1462-1477,
10.1111/nph.15100, 2018.
Miller, B. D., Carter, K. R., Reed, S. C., Wood, T. E., and Cavaleri, M. A.: Only sun-lit leaves of the uppermost
canopy exceed both air temperature and photosynthetic thermal optima in a wet tropical forest, Agricultural and
Forest Meteorology, 301-302, 108347, 10.1016/j.agrformet.2021.108347, 2021.

Oliver, R. J., Mercado, L. M., Sitch, S., Simpson, D., Medlyn, B. E., Lin, Y. S., and Folberth, G. A.: Large but decreasing effect of ozone on the European carbon sink, Biogeosciences, 15, 4245-4269, 10.5194/bg-15-4245-2018, 2018.

Pan, Y., Birdsey, R. A., Fang, J., Houghton, R., Kauppi, P. E., Kurz, W. A., Phillips, O. L., Shvidenko, A., Lewis, S. L., Canadell, J. G., Ciais, P., Jackson, R. B., Pacala, S. W., McGuire, A. D., Piao, S., Rautiainen, A., Sitch, S., and Hayes, D.: A Large and Persistent Carbon Sink in the World's Forests, Science, 333, 988-993, 10.1126/science.1201609, 2011.

Pau, S., Detto, M., Kim, Y., and Still, C. J.: Tropical forest temperature thresholds for gross primary productivity, Ecosphere, 9, e02311, 10.1002/ecs2.2311, 2018.

Poulter, B., MacBean, N., Hartley, A., Khlystova, I., Arino, O., Betts, R., Bontemps, S., Boettcher, M., Brockmann, C., Defourny, P., Hagemann, S., Herold, M., Kirches, G., Lamarche, C., Lederer, D., Ottlé, C., Peters, M., and Peylin, P.: Plant functional type classification for earth system models: results from the European Space Agency's Land Cover Climate Change Initiative, Geosci. Model Dev., 8, 2315-2328, 10.5194/gmd-8-2315-2015, 2015.

Reich, P. B., Sendall, K. M., Stefanski, A., Rich, R. L., Hobbie, S. E., and Montgomery, R. A.: Effects of climate warming on photosynthesis in boreal tree species depend on soil moisture, Nature, 562, 263-267, 10.1038/s41586-018-0582-4, 2018.

Roberts, M. J., Baker, A., Blockley, E. W., Calvert, D., Coward, A., Hewitt, H. T., Jackson, L. C., Kuhlbrodt, T., Mathiot, P., Roberts, C. D., Schiemann, R., Seddon, J., Vannière, B., and Vidale, P. L.: Description of the resolution hierarchy of the global coupled HadGEM3-GC3.1 model as used in CMIP6 HighResMIP experiments, Geosci. Model Dev., 12, 4999-5028, 10.5194/gmd-12-4999-2019, 2019.

Rogers, A., Kumarathunge, D. P., Lombardozzi, D. L., Medlyn, B. E., Serbin, S. P., and Walker, A. P.: Triose phosphate utilization limitation: an unnecessary complexity in terrestrial biosphere model representation of photosynthesis, New Phytologist, 230, 17-22, 10.1111/nph.17092, 2021.

Rogers, A., Medlyn, B. E., Dukes, J. S., Bonan, G., von Caemmerer, S., Dietze, M. C., Kattge, J., Leakey, A. D. B., Mercado, L. M., Niinemets, Ü., Prentice, I. C., Serbin, S. P., Sitch, S., Way, D. A., and Zaehle, S.: A roadmap for improving the representation of photosynthesis in Earth system models, New Phytologist, 213, 22-42, 10.1111/nph.14283, 2017.

Schlesinger, W. H. and Jasechko, S.: Transpiration in the global water cycle, Agricultural and Forest Meteorology, 189-190, 115-117, 10.1016/j.agrformet.2014.01.011, 2014.

Sellar, A. A., Jones, C. G., Mulcahy, J. P., Tang, Y., Yool, A., Wiltshire, A., O'Connor, F. M., Stringer, M., Hill, R., Palmieri, J., Woodward, S., de Mora, L., Kuhlbrodt, T., Rumbold, S. T., Kelley, D. I., Ellis, R., Johnson, C. E., Walton, J., Abraham, N. L., Andrews, M. B., Andrews, T., Archibald, A. T., Berthou, S., Burke, E., Blockley, E., Carslaw, K., Dalvi, M., Edwards, J., Folberth, G. A., Gedney, N., Griffiths, P. T., Harper, A. B., Hendry, M. A., Hewitt, A. J., Johnson, B., Jones, A., Jones, C. D., Keeble, J., Liddicoat, S., Morgenstern, O., Parker, R. J., Predoi, V., Robertson, E., Siahaan, A., Smith, R. S., Swaminathan, R., Woodhouse, M. T., Zeng, G., and Zerroukat, M.: UKESM1: Description and Evaluation of the U.K. Earth System Model, Journal of Advances in Modeling Earth Systems, 11, 4513-4558, 10.1029/2019MS001739, 2019.

Sendall, K. M., Reich, P. B., Zhao, C., Jihua, H., Wei, X., Stefanski, A., Rice, K., Rich, R. L., and Montgomery, R. A.: Acclimation of photosynthetic temperature optima of temperate and boreal tree species in response to experimental forest warming, Global Change Biology, 21, 1342-1357, 10.1111/gcb.12781, 2015.

Shabanov, N., Huang, D., Yang, W., Tan, B., Knyazikhin, Y., Myneni, R., Ahl, D., Gower, S., Huete, A., Aragao, L., and Shimabukuro, Y.: Analysis and Optimization of the MODIS Leaf Area Index Algorithm Retrievals Over Broadleaf Forests, IEEE T. Geosci. Remote, 43, 1855–1865, doi:10.1109/TGRS.2005.852477, 2005.

Slot, M. and Winter, K.: Photosynthetic acclimation to warming in tropical forest tree seedlings, Journal of Experimental Botany, 68, 2275-2284, 10.1093/jxb/erx071, 2017.

Slot, M., Rifai, S. W., and Winter, K.: Photosynthetic plasticity of a tropical tree species, Tabebuia rosea, in response to elevated temperature and $CO_2$, Plant, Cell & Environment, n/a, 10.1111/pce.14049, 2021.

Smith, N. G. and Dukes, J. S.: Plant respiration and photosynthesis in global-scale models: incorporating acclimation to temperature and $CO_2$, Global Change Biology, 19, 45-63, 10.1111/j.1365-2486.2012.02797.x, 2013.

Smith, N. G., Malyshev, S. L., Shevliakova, E., Kattge, J., and Dukes, J. S.: Foliar temperature acclimation reduces simulated carbon sensitivity to climate, Nature Climate Change, 6, 407-411, 10.1038/nclimate2878, 2016.

Spafford, L. and MacDougall, A. H.: Validation of terrestrial biogeochemistry in CMIP6 Earth system models: a review, Geosci. Model Dev., 14, 5863-5889, 10.5194/gmd-14-5863-2021, 2021.

Sullivan, M. J. P., Lewis, S. L., Affum-Baffoe, K., Castilho, C., Costa, F., Sanchez, A. C., Ewango, C. E. N., Hubau, W., Marimon, B., Monteagudo-Mendoza, A., Qie, L., Sonké, B., Martinez, R. V., Baker, T. R., Brienen, R. J. W., Feldpausch, T. R., Galbraith, D., Gloor, M., Malhi, Y., Aiba, S.-I., Alexiades, M. N., Almeida, E. C., de Oliveira, E. A., Dávila, E. Á., Loayza, P. A., Andrade, A., Vieira, S. A., Aragão, L. E. O. C., Araujo-Murakami, A., Arets, E. J. M. M., Arroyo, L., Ashton, P., Aymard C., G., Baccaro, F. B., Banin, L. F., Baraloto, C., Camargo, P. B., Barlow, J., Barroso, J., Bastin, J.-F., Batterman, S. A., Beeckman, H., Begne, S. K., Bennett, A. C., Berenguer, E., Berry, N., Blanc, L., Boeckx, P., Bogaert, J., Bonal, D., Bongers, F., Bradford, M., Brearley, F. Q., Brncic, T., Brown, F., Burban, B., Camargo, J. L., Castro, W., Céron, C., Ribeiro, S. C., Moscoso, V. C., Chave, J., Chezeaux, E., Clark, C. J., de Souza, F. C., Collins, M., Comiskey, J. A., Valverde, F. C., Medina, M. C., da Costa, L., Dančák, M., Dargie, G. C., Davies, S., Cardozo, N. D., de Haulleville, T., de Medeiros, M. B., del Aguila Pasquel, J., Derroire, G., Di Fiore, A., Doucet, J.-L., Dourdain, A., Droissart, V., Duque, L. F., Ekoungoulou, R., Elias, F., Erwin, T., Esquivel-Muelbert, A., Fauset, S., Ferreira, J., Llampazo, G. F., Foli, E., Ford, A., Gilpin, M., Hall, J. S., Hamer, K. C., Hamilton, A. C., Harris, D. J., Hart, T. B., Hédl, R., Herault, B., Herrera, R., Higuchi, N., Hladik, A., Coronado, E. H., Huamantupa-Chuquimaco, I., Huasco, W. H., Jeffery, K. J., Jimenez-Rojas, E., Kalamandeen, M., Djuikouo, M. N. K., Kearsley, E., Umetsu, R. K., Kho, L. K., Killeen, T., Kitayama, K., Klitgaard, B., Koch, A., Labrière, N., Laurance, W., Laurance, S., Leal, M. E., Levesley, A., Lima, A. J. N., Lisingo, J., Lopes, A. P., Lopez-Gonzalez, G., Lovejoy, T., Lovett, J. C., Lowe, R., Magnusson, W. E., Malumbres-Olarte, J., Manzatto, Â. G., Marimon, B. H., Marshall, A. R., Marthews, T., de Almeida Reis, S. M., Maycock, C., Melgaço, K., Mendoza, C., Metali, F., Mihindou, V., Milliken, W., Mitchard, E. T. A., Morandi, P. S., Mossman, H. L., Nagy, L., Nascimento, H., Neill, D., Nilus, R., Vargas, P. N., Palacios, W., Camacho, N. P., Peacock, J., Pendry, C., Peñuela Mora, M. C., Pickavance, G. C., Pipoly, J., Pitman, N., Playfair, M., Poorter, L., Poulsen, J. R., Poulsen, A. D., Preziosi, R., Prieto, A., Primack, R. B., Ramírez-Angulo, H., Reitsma, J., Réjou-Méchain, M., Correa, Z. R., de Sousa, T. R., Bayona, L. R., Roopsind, A., Rudas, A., Rutishauser, E., Abu Salim, K., Salomão, R. P., Schietti, J., Sheil, D., Silva, R. C., Espejo, J. S., Valeria, C. S., Silveira, M., Simo-Droissart, M., Simon, M. F., Singh, J., Soto Shareva, Y. C., Stahl, C., Stropp, J., Sukri, R., Sunderland, T., Svátek, M., Swaine, M. D., Swamy, V., Taedoumg, H., Talbot, J., Taplin, J., Taylor, D., ter Steege, H., Terborgh, J., Thomas, R., Thomas, S. C., Torres-Lezama, A., Umunay, P., Gamarra, L. V., van der Heijden, G., van der Hout, P., van der Meer, P., van Nieuwstadt, M., Verbeeck, H., Vernimmen, R., Vicentini, A., Vieira, I. C. G., Torre, E. V., Vleminckx, J., Vos, V., Wang, O., White, L. J. T., Willcock, S., Woods, J. T., Wortel, V., Young, K., Zagt, R., Zemagho, L., Zuidema, P. A., Zwerts, J. A., and Phillips, O. L.: Long-term thermal sensitivity of Earth's tropical forests, Science, 368, 869-874, 10.1126/science.aaw7578, 2020.

Tramontana, G., Jung, M., Schwalm, C. R., Ichii, K., Camps-Valls, G., Ráduly, B., Reichstein, M., Arain, M. A., Cescatti, A., Kiely, G., Merbold, L., Serrano-Ortiz, P., Sickert, S., Wolf, S., and Papale, D.: Predicting carbon dioxide and energy fluxes across global FLUXNET sites with regression algorithms, Biogeosciences, 13, 4291-4313, 10.5194/bg-13-4291-2016, 2016.

Vårhammar, A., Wallin, G., McLean, C. M., Dusenge, M. E., Medlyn, B. E., Hasper, T. B., Nsabimana, D., and Uddling, J.: Photosynthetic temperature responses of tree species in Rwanda: evidence of pronounced negative effects of high temperature in montane rainforest climax species, New Phytologist, 206, 1000-1012, 10.1111/nph.13291, 2015.

Verhoef, A. and Egea, G.: Modeling plant transpiration under limited soil water: Comparison of different plant and soil hydraulic parameterizations and preliminary implications for their use in land surface models, Agricultural and Forest Meteorology, 191, 22-32, 10.1016/j.agrformet.2014.02.009, 2014.

Vidale, P. L., Egea, G., McGuire, P. C., Todt, M., Peters, W., Müller, O., Balan-Sarojini, B., and Verhoef, A.: On the Treatment of Soil Water Stress in GCM Simulations of Vegetation Physiology, Frontiers in Environmental Science, 9, 10.3389/fenvs.2021.689301, 2021.

Walker, A. P., Beckerman, A. P., Gu, L., Kattge, J., Cernusak, L. A., Domingues, T. F., Scales, J. C., Wohlfahrt, G., Wullschleger, S. D., and Woodward, F. I.: The relationship of leaf photosynthetic traits – Vcmax and Jmax – to leaf nitrogen, leaf phosphorus, and specific leaf area: a meta-analysis and modeling study, Ecology and Evolution, 4, 3218-3235, 10.1002/ece3.1173, 2014.

Walker, A. P., Johnson, A. L., Rogers, A., Anderson, J., Bridges, R. A., Fisher, R. A., Lu, D., Ricciuto, D. M., Serbin, S. P., and Ye, M.: Multi-hypothesis comparison of Farquhar and Collatz photosynthesis models reveals the unexpected influence of empirical assumptions at leaf and global scales, Global Change Biology, 27, 804-822, 10.1111/gcb.15366, 2021.

Way, D. A. and Sage, R. F.: Elevated growth temperatures reduce the carbon gain of black spruce [Picea mariana (Mill.) B.S.P.], Global Change Biology, 14, 624-636, 10.1111/j.1365-2486.2007.01513.x, 2008.

Way, D. A. and Yamori, W.: Thermal acclimation of photosynthesis: on the importance of adjusting our definitions and accounting for thermal acclimation of respiration, Photosynthesis Research, 119, 89-100, 10.1007/s11120-013-9873-7, 2014.

Way, D. A., Stinziano, J. R., Berghoff, H., and Oren, R.: How well do growing season dynamics of photosynthetic capacity correlate with leaf biochemistry and climate fluctuations?, Tree Physiology, 37, 879-888, 10.1093/treephys/tpx086, 2017.

Weedon, G. P., Balsamo, G., Bellouin, N., Gomes, S., Best, M. J., and Viterbo, P.: The WFDEI meteorological forcing data set: WATCH Forcing Data methodology applied to ERA-Interim reanalysis data, Water Resources Research, 50, 7505-7514, 10.1002/2014WR015638, 2014.

Williams, K. D., Copsey, D., Blockley, E. W., Bodas-Salcedo, A., Calvert, D., Comer, R., Davis, P., Graham, T., Hewitt, H. T., Hill, R., Hyder, P., Ineson, S., Johns, T. C., Keen, A. B., Lee, R. W., Megann, A., Milton, S. F., Rae, J. G. L., Roberts, M. J., Scaife, A. A., Schiemann, R., Storkey, D., Thorpe, L., Watterson, I. G., Walters, D. N., West, A., Wood, R. A., Woollings, T., and Xavier, P. K.: The Met Office Global Coupled Model 3.0 and 3.1 (GC3.0 and GC3.1) Configurations, Journal of Advances in Modeling Earth Systems, 10, 357-380, 10.1002/2017MS001115, 2018.

Wilson, K. B., Baldocchi, D. D., and Hanson, P. J.: Leaf age affects the seasonal pattern of photosynthetic capacityand net ecosystem exchange of carbon in a deciduous forest, Plant, Cell & Environment, 24, 571-583, 2001.

Wiltshire, A. J., Duran Rojas, M. C., Edwards, J. M., Gedney, N., Harper, A. B., Hartley, A. J., Hendry, M. A., Robertson, E., and Smout-Day, K.: JULES-GL7: the Global Land configuration of the Joint UK Land Environment Simulator version 7.0 and 7.2, Geosci. Model Dev., 13, 483-505, 10.5194/gmd-13-483-2020, 2020.

Wohlfahrt, G., Bahn, M., Haubner, E., Horak, I., Michaeler, W., Rottmar, K., Tappeiner, U., and Cernusca, A.: Inter-specific variation of the biochemical limitation to photosynthesis and related leaf traits of 30 species from mountain grassland ecosystems under different land use, Plant, Cell & Environment, 22, 1281-1296, 10.1046/j.1365-3040.1999.00479.x, 1999.

Xiao, Z., Liang, S., Wang, J., Xiang, Y., Zhao, X., and Song, J.: Long-Time-Series Global Land Surface Satellite Leaf Area Index Product Derived From MODIS and AVHRR Surface Reflectance, IEEE Transactions on Geoscience and Remote Sensing, 54, 5301-5318, 10.1109/TGRS.2016.2560522, 2016.

Yamaguchi, D. P., Nakaji, T., Hiura, T., and Hikosaka, K.: Effects of seasonal change and experimental warming on the temperature dependence of photosynthesis in the canopy leaves of Quercus serrata, Tree Physiology, 36, 1283-1295, 10.1093/treephys/tpw021, 2016.


Yamori, W., Hikosaka, K., and Way, D. A.: Temperature response of photosynthesis in C3, C4, and CAM
plants: temperature acclimation and temperature adaptation, Photosynthesis Research, 119, 101-117,
10.1007/s11120-013-9874-6, 2014.

Yang, R., Friedl, M. A., and Ni, W.: Parameterization of shortwave radiation fluxes for nonuniform vegetation
canopies in land surface models, Journal of Geophysical Research: Atmospheres, 106, 14275-14286, 2001.

Zeng, Z., Piao, S., Li, L. Z. X., Zhou, L., Ciais, P., Wang, T., Li, Y., Lian, X., Wood, E. F., Friedlingstein, P.,
Mao, J., Estes, L. D., Myneni, Ranga B., Peng, S., Shi, X., Seneviratne, S. I., and Wang, Y.: Climate mitigation
from vegetation biophysical feedbacks during the past three decades, Nature Climate Change, 7, 432-436,
10.1038/nclimate3299, 2017.

Zhang, X. W., Wang, J. R., Ji, M. F., Milne, R. I., Wang, M. H., Liu, J.-Q., Shi, S., Yang, S.-L., and Zhao, C.-
M.: Higher Thermal Acclimation Potential of Respiration but Not Photosynthesis in Two Alpine Picea Taxa in
Contrast to Two Lowland Congeners, PLOS ONE, 10, e0123248, 10.1371/journal.pone.0123248, 2015.

Zhang, Y. and Schaap, M. G.: Weighted recalibration of the Rosetta pedotransfer model with improved
estimates of hydraulic parameter distributions and summary statistics (Rosetta3), Journal of Hydrology, 547, 39-
53, 10.1016/j.jhydrol.2017.01.004, 2017.

Zhao, M. and Running, S. W.: Drought-Induced Reduction in Global Terrestrial Net Primary Production from
2000 Through 2009, Science, 329, 940-943, 10.1126/science.1192666, 2010.

Zhao, M., Running, S. W., and Nemani, R. R.: Sensitivity of Moderate Resolution Imaging Spectroradiometer
(MODIS) terrestrial primary production to the accuracy of meteorological reanalyses, Journal of Geophysical
Research: Biogeosciences, 111, 10.1029/2004JG000004, 2006.

Zhao, M., Heinsch, F. A., Nemani, R. R., and Running, S. W.: Improvements of the MODIS terrestrial gross and
net primary production global data set, Remote Sensing of Environment, 95, 164-176,
10.1016/j.rse.2004.12.011, 2005.

Ziehn, T., Kattge, J., Knorr, W., and Scholze, M.: Improving the predictability of global $CO_2$ assimilation rates
under climate change, Geophysical Research Letters, 38, 10.1029/2011GL047182, 2011.