# Peer review of "Improved representation of plant physiology in the JULES-vn5.6 land surface model: Photosynthesis, stomatal conductance and thermal acclimation"

_Geoscientific Model Development, 2022_

## Referee Comment (RC2)

Comments on "Improved representation of plant physiology in the JULES-vn5.6 land surface model: Photosynthesis, stomatal conductance and thermal acclimation" by Oliver et al. gmd-2022-11

The manuscript reports the addition and changes for the representation of photosynthesis, stomatal conductance and thermal acclimation in the JULES land surface model by adapting the photosynthesis model, the stomatal conductance model and adding the thermal acclimation of the photosynthetic capacity. It further evaluated the impacts of these changes on carbon, energy and water fluxes by comparing the simulation results against other available estimates of GPP from FluxCom and MODIS, turbulent heat fluxes and evaporative fraction (LE, H, EF=LE/(LE+H)) from FluxCom and ET from FluxCom and GLEAM. The description and argumentation for the improvements are in general clearly stated. After studying the manuscript, I wish to point out the following technical issues for considerations by the authors.

1. Because the used references are estimates themselves, it is rather difficult to ascertain if a better agreement with these references reflects a better representation of the physical processes or a better fitting due to optimized parameters. One notices the fine differences in certain parameters e.g. *Vcmax* in Table 2 and wonders how the simulation results will differ if these parameters are used on a site by site evaluation and the results evaluated against the site observations (instead of the reference estimates). As a minimum, the authors can add such information in the supplementary material and add a short discussion.

2. The treatment of the soil water stress is through its impact on the net photosynthesis in eq. 1b. This is likely not what happens on the process level. Wang et al. (2021, GMD) demonstrated that the soil water stress should be linked to leaf water stress. It is not reasonable to ask the authors to redo all the simulations for different treatments of the soil water stress, but a site by site comparison should reveal the effectiveness of each treatment.

3. What is the meaning of $\theta$ in eq. 4?

4. It is not always clearly stated what time step was used in calculating the relative RMSE with eq. 11 for the different evaluations.

5. The authors report (*Vcmax*, *Jmax* and *Jmax:Vcmax*) but it is not clear why a third quantity is needed while one can be derived it from any other two.

6. It is cosmetic, but to this reviewer the plots in Fig. 4 visualize better if they are rotated by 90 degrees.

7. The authors stated the overestimation of ET in SON (Fig. 3b) by all model configurations but a short discussion to the reasons should be provided.

---

## Author Comment (AC1)

**Reply to the reviewers' comments: gmd-2022-11**
R. J. Oliver *et al*.
**Correspondence:** rfu@ceh.ac.uk

We would like to thank the three reviewers for their time to read and comment on this manuscript. We have addressed all the revisions, and hope this improves the manuscript satisfactorily. Blue text below is our response to the reviewers' comments (reproduced in black).

**Reviewer 1**

The preprint manuscript gmd-2022-11 "Improved representation of plant physiology in the JULES-vn5.6 land surface model: Photosynthesis, stomatal conductance and thermal acclimation" presents the implementation and parametrisation of three additional photosynthesis and stomatal conductance formulations (Farquhar, Medlyn and "AcKK" in addition to original Collatz formulation) into the JULES LSM. Parametrisations are mostly based on large datasets or pre-existing peer-reviewed work. The four model configurations were analysed on site level with data from 17 EC sites as well as globally using drivers from the WFDEI meteorological dataset. An additional RCP8.5 scenario simulation was also performed and evaluated.

I found this manuscript to be an interesting read and I do not have any major revision requests. The results are (mostly) clearly presented, properly discussed and the conclusions are accurately drawn. While the presented model modifications are not new, they provide new insights in model behaviour and expand the JULES model functionality. Additionally, the new model configurations also provide a possibility for the model to be run in an "ensemble" mode (similar in style to CMIP) that would allow e.g., improved detection of model weaknesses and uncertainties related to processes and variable estimates. One thing that could be discussed is the effect of static PFT's to the simulations and results.

Minor revision requests:

L167        The last reference Mercado et al. – 'et al' is in italics and missing point.

Done

L181        I believe this should be GC3.1 instead of GC.1.

The reviewer is right, many thanks for pointing this out, it has been changed to GC3.1.

L200-202        So soil water stress is a linear response function (as in Best et al. 2011), but are the parameters fixed throughout the different biomes/soil types? I think it is reasonable to exclude possible $\beta$-function modifications (such as sigmoidal response or $\beta^s$ formulations) from these experiments, but I would like a clarification to the manuscript, i.e., just stating that the parameters are fixed or that they depend on the soil type is ok.

We have added the following text (Line 201): "The critical and wilting point soil moisture concentrations vary by soil type in these simulations.".

We have added more information about how the soil properties are prescribed at Line 395: "Prescribed parameters were used for the hydraulic and thermal properties of the soil from a modified version of the H1 lookup-table from Zhang & Schaap (2017) that depends upon the soil textural type from SoilGrids (Hengl *et al.* 2014)."

L239        You have here an undefined parameter $\theta$ and I think you mean $\alpha$.

θ is a curvature factor, we have added the following to the text to clarify (Line 242): "…and $\theta$ a non-rectangular hyperbola smoothing parameter which takes a value of 0.9 (unitless) following Medlyn et al. (2002)."

L247          Just a suggestion for the last 'large' division – add normal brackets around both numerator and divider and add '-1' exponent to the latter and write them as a product. If you like the original formulation more just use that.

Thank you for the suggestion, we prefer the original formulation as this tends to be more commonly used (e.g. Eqn 1 in Kattge & Knorr 2007; Eqn 7 in Kumarathunge *et al*., 2019).

Kattge & Knorr (2007). Temperature acclimation in a biochemical model of photosynthesis: a reanalysis of data from 36 species. Plant, Cell & Environment, 30, 1176-1190.

Kumarathunge et al., (2019). Acclimation and adaptation components of the temperature dependence of plant photosynthesis at the global scale. New Phytologist, 222, 768-784

L310-312      Although, fitting $g_1$ to $g_s$ is not the most complicated of tasks, at least some more information should be given about this. I think you are using nonlinear regression, but did you estimate the goodness of the fit as in Lin et al. (2015) or use a separate validation set to control for overfitting? Regardless, the dataset seems heterogenous enough to prevent overfitting by itself.

We have added the following text for clarity (Line 320): "A non-linear mixed-effects model was used to estimate the model slope coefficient, *g1*, for each PFT, where individual species were assumed to be the random effect to account for the differences in the *g1* slope among species within the same group, following Lin et al. (2015)."

L346          Is the mean seasonal cycle produced over entire measurement period or do you have one average cycle for each year? With former you are essentially smoothing the timeseries and reducing year-to-year variability which makes sense from a "climatic" perspective, but you might miss interannual variability. Although, since you are not calibrating/optimising model parameters this is not really an issue. Additionally, could you add time-intervals (which years of measurements you have used) to Table S1?

We have clarified this by adding to the text (Line 363): "The mean seasonal cycle calculated over the entire measurement period is used in order to assess the mean model behaviour."

We have added information about the time period used in simulations to Table S1.

L385          You should explain what conservative here means (flux preserving) or give a reference.

We have added the following text (Line 417): "…which ensures the physical conservation of each variable."

L393-396      Just to clarify if I understood this correctly, you generate a separate spin-up for each model configurations and the difference of differences is taking "Collatz" as the baseline configuration for site simulations?

That is correct, each model configuration has its own spin-up, consequently the transient simulations start from different points. To assess the change in GPP, LE and H over the simulation period due to the addition of a new process alone (and not due to different initial starting points) we use the difference of difference approach. For the baseline this uses the simulation in which everything else is identical apart from the process of interest, so for instance, to look at the impact of changing photosynthesis schemes, the baseline (*Y*, without the process) is Clz.Jac and the simulation with the process of interest (*X*) is Fq.Jac. To look at the impact of the stomatal conductance schemes *Y*=Fq.Jac, *X*=Fq.Med, to look at the impact of acclimation *Y*=Fq.Med, *X*=AcKK.Med (here, both are using Med stomatal conductance, and Farquhar photosynthesis, but *X* has the addition of thermal acclimation).

We explain this in the legend of Figure 7, but have clarified in the text also at Line 434: "For example, to look at the impact of changing photosynthesis schemes, $X$ = Fq.Jac and $Y$ = Clz.Jac. In this case, both configurations are using the Jacobs $g_s$ scheme, only the photosynthesis scheme changes from Collatz to Farquhar. The impact of changing $g_s$ scheme is assessed where $X$ = Fq.Med and $Y$ = Fq.Jac. The impact of thermal acclimation is assessed where $X$ = AcKK.Med and $Y$ = Fq.Med, here both simulations use the Farquhar photosynthesis scheme and the Medlyn $g_s$ scheme, but $X$ has the addition of thermal acclimation of photosynthesis."

L398        You have a different realization of a process, not a complete lack of one, so I would suggest rephrasing this sentence.

We have changed the text (Line 431): "where $X$ represents the simulation with the process of interest and $Y$ represents the simulation with the alternative representation."

L419        "Jmax:Vcmax" ratio has ":" in the index (or is this deliberate as it appears like this elsewhere in the paper as well).

This is deliberate as it is the ratio, but we have changed the ":" from subscript to normal to be consistent throughout the manuscript.

L468        Because you have RMSE in the panel headings, I was initially reading this as "negative values are improvements" (since RMSE would be smaller). If you want to "fix" this, change the order in EQ 12 to "new – old".

In Figure 1 we summarise the changes in RMSE by showing the relative improvement for each model configuration compared to the Clz.Jac baseline. Plotting the 'relative RMSE' was a more effective way to clearly show the changes in RMSE, rather than the RMSE values themselves. This is calculated as in equation 11 and is calculated so that positive values show an improvement in the model. We have clarified these points at Line 365 and in the legend of Figure 1, and in the title of the Figure 1 plots.

L481        Figure 2 headings are now telling a different story. Here you state that comparisons are to the original, but in the caption it is between the different configurations (based on headings you are not showing differences between Fq.Jac and AcKK.Med as well as Fq.Med and Clz.Jac.). Please change either image or text to what you want to show.

Apologies, this was not very clear. We have changed the text (Line 519): "The impact of changes in the photosynthesis scheme, $g_s$ scheme, adding thermal acclimation of photosynthetic capacity and the overall change on simulated GPP, LE and H are shown in Figure 2 by comparing each of the new JULES configurations to the configuration with the alternative process representation." We have also changed the legend for Figure 2 to make this clear.

L612        It is a bit difficult to properly interpret the RMSE values from S8 and S9 so I wanted to ask, how clearly defined are these areas? In many places the differences seem quite minimal (and I don't think there's a good way to improve the images).

The differences in RMSE between the observations and each of the different model configurations are typically small, we state this in Line 631. The coloured bars in Figure 4 and Figure 5 correspond to which model configuration gives the lowest RMSE compared to observations by region and season, and we show the actual RMSE values in Tables S4-S7. Figure 6 then shows the spatial pattern i.e. which model configuration compares best to observations (has the lowest RMSE) by season. Figs S8 and S9 (now Figs. S12 and S13) support this by showing the actual RMSE values of each model configuration compared to the different observational products.

L706        I believe there was no competition among species or changing PFT zones/percentages (this came to mind with the mention of boreal region here)? This is one thing you might want to mention or speculate on somewhere in the discussion (maybe in or after the last paragraph under this heading).

We have added the following paragraph to discuss the implications of running JULES with vegetation dynamics enabled (Line 850): "The simulations presented in this work use a prescribed map of vegetation cover which means the extent and location of each PFT does not change over time. The model can alternatively be run with dynamic vegetation enabled, which means the model predicts the extent of each PFT, and therefore vegetation cover can change in space and time as PFTs compete with each other in response to changing climatic conditions. Yet to be explored as part of this work, is how changes to the plant physiology routines, as implemented here, might affect the extent of different PFTs over time when vegetation dynamics is enabled. For example, changes to the temperature response of photosynthesis may lead to a competitive advantage of one PFT over another, and therefore the vegetation distribution may be very different as temperatures rise compared to simulations that either use the original Collatz temperature sensitivities or do not include thermal acclimation of photosynthesis. We hypothesise, for example, that allowing thermal acclimation of the temperature sensitivity of photosynthesis would make the vegetation distribution more stable in a warmer climate as vegetation can adjust its photosynthetic capacity to function more efficiently as temperatures rise. Applied in a coupled ESM, a change in vegetation distribution would impact projections of future climate change."

L712        Missing space after bracket.

Done

L890        You use two types of doi formats (long and short), and I would suggest sticking with just one.

Done. We have changed all references to use the short format.

Fig. S2        is not referenced in the manuscript or supplementary material.

We now refer to Fig. S2 in Lines 354 and 355.

**Reviewer 2**

The manuscript reports the addition and changes for the representation of photosynthesis, stomatal conductance and thermal acclimation in the JULES land surface model by adapting the photosynthesis model, the stomatal conductance model and adding the thermal acclimation of the photosynthetic capacity. It further evaluated the impacts of these changes on carbon, energy and water fluxes by comparing the simulation results against other available estimates of GPP from FluxCom and MODIS, turbulent heat fluxes and evaporative fraction (LE, H, EF=LE/(LE+H)) from FluxCom and ET from FluxCom and GLEAM. The description and argumentation for the improvements are in general clearly stated. After studying the manuscript, I wish to point out the following technical issues for considerations by the authors.

1. Because the used references are estimates themselves, it is rather difficult to ascertain if a better agreement with these references reflects a better representation of the physical processes or a better fitting due to optimized parameters. One notices the fine differences in certain parameters e.g. Vcmax in Table 2 and wonders how the simulation results will differ if these parameters are used on a site by site evaluation and the results evaluated against the site observations (instead of the reference estimates). As a minimum, the authors can add such information in the supplementary material and add a short discussion.

For this work we used a data-driven approach to parameterise the model. New parameters for each model configuration are adopted from recent large observational datasets that synthesise global experimental data, we therefore do not fit any parameters (with the exception of the temperature sensitivity for the C3 grass PFT). For example, the temperature sensitivity parameters for the Farquhar model are from the work of Kumarathunge $et$ $al.$, (2019), the estimates of $V_{cmax25}$ and $J_{max25}$ are from the database of Walker $et$ $al.$, (2014), and the $g_1$ values are from the database of Lin $et$ $al.$, (2015). There are only fine differences in the estimates of $V_{cmax25}$ between the Collatz and Farquhar implementations, but it was necessary to find these new estimates as part of this work to ensure that we were using $V_{cmax25}$ values that had been derived using the Farquhar model to ensure consistency between data and model as has been shown to be important by Walker $et$ $al.$, (2021)

For the site level simulations, we calculate the RMSE of the mean seasonal diurnal cycle for each model configuration compared to the observations of GPP and ET from FLUXNET. Then following Eq 11 we calculate the relative RMSE compared to the Clz.Jac baseline. Summarising the changes in RMSE in this way (i.e. showing the relative RMSE in Figure 1) was a more effective way to clearly show the changes in RMSE, rather than the RMSE values themselves, and was also a way to summarise a lot of information from the seasonal diurnal cycles simulated at each site by each model configuration. The statistic is calculated so that positive values show an improvement in the model compared to the Clz.Jac baseline, but this also means an improvement compared to the Fluxnet observations. We have clarified these points at Line 365 and in the legend of Figure 1, and in the title of the Figure 1 plots. We therefore do perform evaluation against the observations at each of the 17 eddy covariance sites, but to synthesise the results we plot the relative RMSE in Figure 1. However, we fully appreciate that being able to see the fit to observations for each model configuration at each site is valuable. Therefore, as the reviewer suggests, we have added Fig. S5 and Fig. S6 to show the mean seasonal (all seasons) diurnal cycles for GPP and EF respectively, and have added to these plots the calculated RMSE for each model configuration compared to observations. It is these data that are summarised (for MAM and JJA) in Figure 1 in the main manuscript. The discussion (section 4.1 Site level evaluation) does discuss the results in the context of each site and how they compare to observations, so to this we have added references to Fig. S5 and Fig. S6 which greatly helps the clarity of the discussion.

Kumarathunge et al., (2019). Acclimation and adaptation components of the temperature dependence of plant photosynthesis at the global scale. New Phytologist, 222, 768-784

Walker et al., (2014). The relationship of leaf photosynthetic traits – Vcmax and Jmax – to leaf nitrogen, leaf phosphorus, and specific leaf area: a meta-analysis and modeling study, Ecology and Evolution, 4, 3218-3235, 10.1002/ece3.1173.

Lin et al., (2015). Optimal stomatal behaviour around the world, Nature Climate Change, 5, 459-464, 10.1038/nclimate2550.

Walker et al., (2021). Multi-hypothesis comparison of Farquhar and Collatz photosynthesis models reveals the unexpected influence of empirical assumptions at leaf and global scales, Global Change Biology, 27, 804-822, 10.1111/gcb.15366.

2. The treatment of the soil water stress is through its impact on the net photosynthesis in eq. 1b. This is likely not what happens on the process level. Wang et al. (2021, GMD) demonstrated that the soil water stress should be linked to leaf water stress. It is not reasonable to ask the authors to redo all the simulations for different treatments of the soil water stress, but a site by site comparison should reveal the effectiveness of each treatment.

We agree that the treatment of soil moisture stress is very important and will have a large impact on modelled carbon and water fluxes. How soil moisture stress is represented within JULES, however, is beyond the scope of this current work. Nevertheless, the reviewer has raised an important point that we discuss further in section 5.4 Lines 863:

"The treatment of soil moisture stress in JULES is through a linear response function (the $\beta$ function, Eq. 12 in Best et al., 2011), the use of which in JULES and other LSMs has been identified as a key source of uncertainty (Blyth et al., 2011; Verhoef & Egea, 2014; Vidale et al., 2021). Incorrect representation of soil moisture stress has large impacts for modelled carbon and water fluxes, and is of particular importance as droughts are predicted to increase in frequency or intensity in the future. Work is ongoing to improve the representation of soil moisture stress in JULES. Harper et al., (2021) investigated alternative parameterisations for $\beta$ and found that increasing modelled soil depth and therefore plant access to deep soil moisture improved the simulation of soil moisture stress at eddy covariance flux tower sites. In addition, using soil matric potential instead of volumetric water content in the $\beta$ function allowed for PFT specific parameterisation of soil moisture stress responses to further improve modelled fluxes. Vidale et al., (2021) explored combinations of non-linear $\beta$ function responses applied at different points in the photosynthesis – $g_s$ pathway (i.e. carbon assimilation, $g_s$, or mesophyll conductance). They found that treatments allowing $\beta$ to act on vegetation fluxes via stomatal and mesophyll routes were able to better capture the spatiotemporal variability in water use efficiency during the growing season. However, in addition to these alternative parameterisations of $\beta$, further developments to how the soil-plant hydraulic system is represented in JULES are being made, including an optimality based plant hydraulic transport model recently implemented in JULES (Eller et al., 2020)."

3. What is the meaning of $\theta\theta$ in eq. 4?

We have added the following to the manuscript (Line 242): "…..and $\theta$ a non-rectangular hyperbola smoothing parameter which takes a value of 0.9 (unitless) following Medlyn et al. (2002)".

4. It is not always clearly stated what time step was used in calculating the relative RMSE with eq. 11 for the different evaluations.

We have updated the text to clarify this, Line 365: "We evaluate the site-level simulations with RMSE (root mean square error) for the seasonal diurnal cycle of simulated (daytime) fluxes (GPP and EF). For each site, the time period of the simulation and therefore evaluation period is stated in Table S1. We summarise the changes in RMSE using the relative improvement for each model configuration ($i$) compared to the current standard JULES configuration of Collatz with Jacobs (Clz.Jac). The statistic is calculated so that positive values show an improvement compared to Clz.Jac and therefore a better comparison to the observations."

5. The authors report (Vcmax, Jmax and Jmax:Vcmax) but it is not clear why a third quantity is needed while one can be derived it from any other two.

The $J_{max}:V_{cmax}$ ratio is an important parameter for the thermal acclimation model. It can be derived from just reporting $V_{cmax}$ and $J_{max}$, but for clarity we show it in Table 2.

6. It is cosmetic, but to this reviewer the plots in Fig. 4 visualize better if they are rotated by 90 degrees.

We have updated the plots as suggested.

7. The authors stated the overestimation of ET in SON (Fig. 3b) by all model configurations but a short discussion to the reasons should be provided.

We discuss this point in section 5.4, but as suggested by the reviewer we have expanded our discussion (Line 825): "For example, in SON the high GPP and ET bias occurs in the northern temperate and boreal region which could be linked to a lack of photosynthetic phenology in the model. Towards the end of the growing season leaves in this region have reduced nitrogen content and therefore lower photosynthetic capacity, but because it uses a fixed value for photosynthetic capacity JULES maintains a high rate of carbon assimilation despite having seasonal LAI."

**Reviewer 3**

The preprint manuscript gmd-2022-11 "Improved representation of plant physiology in the JULES-vn5.6 land surface model: Photosynthesis, stomatal conductance, and thermal acclimation" focused on improving the representation of photosynthesis and stomatal conductance parameterization within the JULES land surface model. This work is interesting and I think the paper should be published as it brings new and useful information to the scientific literature.

Minor revision:

Line 202-206: The multi-layer canopy module can deal with the distribution of radiation and energy in vegetation very well. However, there is a hypothesis here that the parameterization of vegetation canopy needs accurate characterization, which is recommended to be described in a little detail.

We thank the reviewer for raising this point, and have added a discussion to the manuscript that justifies the options used in this study for the canopy radiation scheme (Line 206): "The implementation of a multilayer canopy for light interception in JULES was shown to improve modelled canopy scale photosynthetic fluxes at eddy covariance sites compared to the 'big leaf approach' (Jogireedy et al., 2006, Mercado et al., 2007, Blyth et al 2011). Specifically, the multi-layer approach better captured the light response and diurnal cycles of canopy photosynthesis. While light inhibition of leaf respiration and changing photosynthetic capacity with canopy depth are supported by observations (Meir et al., 2002; Atkin et al., 1998; Atkin et al., 2000). Sunfleck penetration through the canopy and the differential effects of direct and diffuse beam radiation on modelled carbon and water exchange in JULES were studied by Mercado et al., (2009). This enabled JULES to reproduce the different light-response curves of GPP under diffuse and direct radiation conditions at both a broadleaf and needleleaf temperate forest."

Line 239: what is $\theta$?

Text has been added (Line 242): "…and $\theta$ a non-rectangular hyperbola smoothing parameter which takes a value of 0.9 (unitless) following Medlyn et al. (2002)."

Line 241: it is suggested to show the Q10 functions here or in the supplementary.

We have added a description of the Collatz $Q10$ functions to the supplementary as Notes S1, and updated the main manuscript as follows (Line 250): "JULES currently uses $Q_{10}$ functions in the Collatz scheme to describe the temperature dependency of $V_{cmax}$, $K_c$, $K_o$, and $\Gamma$ (see Notes S1). In our implementation of the Farquhar scheme, temperature sensitivities for the $K_c$, $K_o$, and $\Gamma$ are taken from…".

Line 249 and 296: the unit of the gas constant R, 8.314 J·mol-1K-1, suggest using the same expression.

This has been updated.

Line 411-424 and 452-462: How long is the simulation period for these sites? From Figure 1, different schemes have significant differences in the simulation performance of GPP and EF at different sites. Is it due to insufficient accumulation, especially for EBF/BET-tr and NET?

We have added information regarding the simulation period for each site to Table S1. The length of available data to use for simulation and evaluation at each site does vary substantially. This may contribute to the range of performance at different sites, however more influential will be how well the model is capturing things such as leaf area index, and soil moisture dynamics at each site which both help determine modelled photosynthesis and $g_s$. For example, site-level observations of leaf area index were only available for three of the sites, all other sites used the JULES phenology scheme to simulate leaf area index which will introduce differences between sites depending on how well the phenology scheme behaves, but this was beyond the scope of our study to evaluate.

Line 522: Could you please show the results of GPP, LE, and H from the original JULES model? Then add the difference between each new configuration to the original one.

We have added plots of the absolute mean GPP, LE and H simulated in both JJA and DJF by each model configuration to the SI (Fig. S8 and Fig. S9).

Line 599-601: The tropical forests appeared not only in Amazon and central Africa but also in Southeast Asia, including South China, Indo-China, Malay Peninsula, and regions to their south. It seems that AcKK.Med is not the only one who shows the best in the tropical forest.

We have updated the manuscript as follows (Line 632): "Figure 6a & b show that in the tropical forests of the Amazon basin, central Africa and Southeast Asia (Indonesia, Papua New Guinea, Malaysia), in both JJA and DJF (Fig. S7a & b for DJF), GPP simulated including thermal acclimation (AcKK.Med) compares best to both FluxCom and MOD17 across large spatially consistent areas. Outside of these areas, Fq.Jac also improves the simulation of GPP in the tropics, as does the Medlyn $g_s$ model (Fq.Med) in JJA in South China and Indo-China."

Line 753-759: The reviewer agreed the understanding of tropical forests is still lacking. The complexity of the canopy process is not well handled in the current model, which is also one of the problems restricting the JULES model and other ESMs. It is suggested that the author consider adding some discussion from this aspect.

We agree with the reviewer that this is an important to consider and have added discussion around the lack of complex canopy processes represented in JULES and other LSMs generally in section 5.4 at Line 879: "Whilst the development of multi-layer canopy radiation models in LSMs has improved the simulation of radiation and energy within vegetation canopies, the interception of light by plants in JULES, like most LSMs, is not well represented despite being critical to predicting the uptake of carbon by plants (Loew et al., 2014). LSMs generally make the simplifying assumption that leaves are randomly arranged in space, instead of being clustered into tree crowns or around branches, leaving gaps in and around the canopy. Shortwave radiation is used by plants to photosynthesise, and canopy structure has a direct impact on the fraction of this radiation absorbed, therefore canopy architecture plays an important role in the partitioning of incident solar radiation, photosynthesis, transpiration and momentum fluxes (Braghiere et al., 2019). More recently, alternative approaches are being considered to represent the forest light environment in LSMs to account for the structural effects of vegetation on radiation partitioning, ranging from canopy clumping parameterisations (Braghiere et al., 2020; Braghiere et al., 2021) to 3-dimensional models of the canopy light environment (Kobayshi et al., 2012; Hogan et al., 2018), embedded in radiative transfer schemes, although the latter tend to be computationally expensive (Yang et al., 2001). Braghiere et al., 2019 incorporated canopy clumping from satellite data into JULES which resulted in an increase in carbon uptake by photosynthesis. The greatest effect were in the tropics, where the canopy clumping parameterisation allowed more light to reach the lower layers of the canopy where photosynthesis tends to be limited by light availability."

Line 856: It is suggested that authors could share data and scripts in a similar opensource way. MOSRS registration takes too long to examine and verify, so there is no way to review it more deeply.

We apologise for this and completely understand that registration with MOSRS can be both time consuming and generally just a pain. I think the time between applying for a MOSRS account and it being approved is not much shorter, when this manuscript was submitted it unfortunately coincided with an interruption to the MOSRS account granting system. It is very unfortunate that we cannot share the JULES code via zenodo or any other similar easy to access repositories as this violates the JULES user licence agreement.

Table S1: What is the period corresponding to the observation data of each station? And it is suggested to show the prescribed LAI value of each station.

We have added the simulation period used to Table S1, and this therefore corresponds to the observation period used for evaluation. We have indicated which sites use prescribed LAI from site data (which is available from Fluxnet), but haven't shown the value as this is a time series of data.

After Figure S12: What is the purpose of this document? Restrepo-Coupe, N, 2013.

This is a reference for the LBA sites used in the site-level evaluation – we have added the heading 'References' to clarify.